# A Faster Generalized Two-Stage Approximate Top-K

**Yashas Samaga**                                                    *syashas@cs.washington.edu*
*University of Washington, Seattle*
*(work done while at Google DeepMind)*

**Varun Yerram**                                                            *y.varun@nyu.edu*
*New York University*
*(work done while at Google DeepMind)*

**Spandana Raj Babbula**                                              *sbabbula@google.com*
*Google DeepMind*

**Prateek Jain**                                                          *prajain@google.com*
*Google DeepMind*

**Praneeth Netrapalli**                                              *pnetrapalli@google.com*
*Google DeepMind*

**Reviewed on OpenReview:** *https://openreview.net/forum?id=izqZ1Crpjz*

## Abstract

We consider the Top-$K$ selection problem, which aims to identify the largest $K$ elements in an array. Top-$K$ selection arises in many machine learning algorithms and often becomes a bottleneck on accelerators, which are optimized for dense matrix multiplications. To address this problem, Chern et al. (2022) proposed a fast two-stage *approximate* Top-$K$ algorithm that: (i) partitions the input array into equal-sized chunks and selects the top-1 element from each partition; and (ii) sorts the resulting *smaller subset* and returns the top $K$ elements. In this paper, we generalize the first stage so that each partition selects the top $K'$ elements (for $1 \leq K' \leq K$). Our contributions include: (i) an expression for the expected recall of this generalized algorithm under random partitioning, and a demonstration that choosing $K' > 1$ with *fewer partitions* in the first stage more effectively reduces the input size to the second stage while maintaining the same expected recall as the original algorithm; (ii) a bound on the expected recall of the original algorithm as a function of the algorithm parameters that is provably tighter by a factor of 2 than the bound reported by Chern et al. (2022); and (iii) an implementation of our algorithm on Cloud TPUv5e that achieves approximately an order of magnitude speedup over the original algorithm without sacrificing recall.

## 1 Introduction

Identifying the top-K elements in an array is an essential building block of many algorithms. Beyond its common applications in Maximum Inner Product Search (MIPS) or K-Nearest-Neighbors (KNN) (Chern et al., 2022; Li et al., 2023), it has recently become important for optimizing training and inference of large foundation models. Large language models (LLMs) use the Top-$K$ operation to exploit the sparsity in several components, including classification logits (Samaga B L et al., 2024), MLP blocks (Liu et al., 2023; Samaga B L et al., 2024; Alizadeh et al., 2024), attention mechanisms (Roy et al., 2021; Madaan et al., 2023; Deshmukh et al., 2025; Synk et al., 2025) and KVCache compression (Behnam et al., 2025; Yang et al., 2025; Tang et al., 2024). It is also used in retrieval augmented generation (Lewis et al., 2021; Borgeaud et al., 2022), sampling tokens (Shen et al., 2024), mixture-of-experts (Dai et al., 2024; He, 2024), and accelerating distributed training (Shi et al., 2019; Ruan et al., 2023).

Given the scale of these models, their training and inference are typically performed on accelerators such as TPUs and GPUs. However, computing Top-$K$ on these devices can be expensive. On TPUv5e and A100, finding the top-2% of the hidden activations of Gemma 2 9B's (Team et al., 2024) feedforward blocks[1] during training using `jax.lax.top_k` takes $27\times$ and $4.8\times$ longer, respectively, than the corresponding matrix multiplication that generated those activations. Ideally, computing Top-$K$ should be negligible compared to the matrix multiplications.

As a workaround, foundation models increasingly use *approximate* Top-K algorithms, which are generally robust to these approximations (Samaga B L et al., 2024; Key et al., 2024).

Chern et al. (2022) introduced a hardware-friendly approximate Top-$K$ algorithm that operates in *two stages*. In the first stage, the input array is divided into a fixed number of equal-sized buckets, and the top-1 element from each bucket is selected. In the second stage, these top-1 elements are sorted, and the first $K$ elements are returned. They quantify the approximation error in terms of *expected recall* (Wang et al., 2014), defined as the proportion of the actual top-K elements retrieved in the first stage averaged over all permutations of the inputs. They derive a closed-form expression that relates the expected recall to the number of buckets, which is then used to *choose a number of buckets* that is sufficient to maintain a *user-specified average recall target*. We refer to this method as *the original algorithm*.

At the time of writing, this algorithm was implemented in `jax.lax.approx_max_k` for TPUs. In the earlier example of finding the top-2% of the hidden activations, this algorithm with a recall target of 95% still takes $9\times$ more time than the matrix multiplication on TPUv5e.

As we explain in Sections 2 and 6, the second stage is typically the bottleneck, since the first-stage computation is relatively inexpensive and efficiently parallelizable. In fact, in tasks that require finding the top-K elements in each column or row of a matrix product, the first stage can be "fused"[2] (Snider & Liang, 2023) with the matrix multiplication, effectively hiding much of its cost. Therefore, improving the efficiency of this algorithm requires reducing the number of elements sorted in the second stage without sacrificing the expected recall, while ensuring that the first-stage computation remains inexpensive. Our main contribution is a new algorithm that achieves this, which we describe next.

We accomplish this by generalizing the first stage of the approximate Top-$K$ algorithm of Chern et al. (2022) to select top-$K'$ elements from each bucket (where $K' < K$) instead of restricting selection to the top-1. This increases the total number of elements sorted in the second stage to $B \cdot K'$, where $B$ is the number of buckets. However, *our key result shows that for a large set of values of $K$, array size $N$ and recall targets, it is possible to reduce the number of buckets $B$ sufficiently that the smallest number of elements to sort in the second stage $(B \cdot K')$ is achieved by some $K' > 1$ while ensuring that the first stage does not become the bottleneck.*

**Theoretically**, we derive an expression relating $K'$, $K$ and $B$ to the expected recall. Using this expression, we select the parameters $K'$ and $B$ for our algorithm that satisfy the user-specified average recall target. While the full potential is realized by choosing $K' > 1$, interestingly, even for $K' = 1$, the setting of Chern et al. (2022), our bound on the number of buckets is provably a factor of 2 tighter than theirs, which improves parameter selection and, in most cases, more than doubles the performance of the original algorithm. For $K' > 1$, the gains are even higher.

**Empirically**, we implement our generalized algorithm $(K' > 1)$ for TPUs using Pallas and demonstrate an order of magnitude speedup in latency on TPUv5e chips compared to the original algorithm. We provide two implementations: (i) an unfused version that executes the two stages as two separate *kernels* (background on "kernels" is in Section 2); and (ii) a matmul-fused version that fuses the first stage with a matrix multiplication (background on fusion and related matmuls are in Sections 2 and 3). In the earlier example of identifying the top 2% of hidden activations, our implementations for $K' = 4$ make the Top-$K$ step $24\times$

---

[1]This involves an einsum contraction between a 3D tensor of shape `[batch_size, seqlen, model_dims]` and a 2D weight matrix of shape `[model_dims, hidden_dims]`, contracting along the `model_dims` axis (i.e., `"bsm,mh -> bsh"`). Top-K is then applied along the `hidden_dims` axis.

[2]In this context, *fusion* refers to merging the top-K selection logic with the matrix multiplication logic into a single fused operation such that the top-K step can, in some cases, be performed at no additional cost. We discuss this in more detail in Section 2.

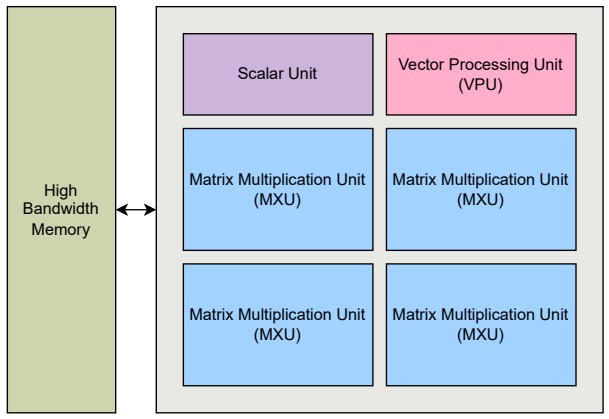

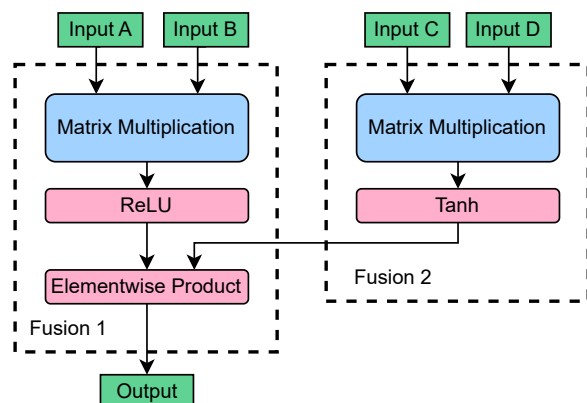

Figure 1: **Overview of TPUv5e subsystems.** The TPUv5e chip features four Matrix Multiplication Units (MXUs) dedicated to matrix-matrix multiplications along with a Vector Processing Unit (VPU) that performs general vector operations, such as activations and softmax. The chip also includes a scalar unit for calculating memory addresses, managing control flow, and performing similar tasks.

Figure 2: **Decomposing a program into smaller subprograms.** In this example, a program has been broken down into two subprograms, each containing several elementary operations. The subprograms are executed in an order that satisfies their dependencies, with Fusion 2 executing before Fusion 1.

faster than `jax.lax.approx_max_k`. This reduces the time taken by approximate Top-$K$ to below that of the corresponding matrix multiplication, resulting in an overall speedup of $6.7\times$. These localized gains translate into significant end-to-end gains; in Appendix A.13, we provide a holistic breakdown of the training costs of a sparse transformer block. We share the code for our TPU implementations in Appendix A.8 and A.9.

## 2 Background

### 2.1 Organization of compute on accelerators

Compute resources in most accelerators are distributed across several distinct subsystems, each specialized for different types of operations (see Figure 1). On TPUs, the vast majority of compute is concentrated in two subsystems: the Matrix Multiply Unit (MXU) and the Vector Processing Unit (VPU) (Norrie et al., 2020; Google Cloud, 2024b). MXUs account for more than 95% of compute FLOPS (Chern et al., 2022; Google Cloud, 2024b); consequently, only MXU-bound programs reach peak FLOPS utilization. Similarly, on GPUs, compute resources are primarily spread across two subsystems: TensorCores for matrix multiplications and CUDA cores for scalar/vector computations. As with TPUs, most FLOPS are concentrated in TensorCores (NVIDIA, 2024).

### 2.2 Kernels and fusions

Programs for accelerators are typically decomposed into a series of smaller subprograms, known as *kernels*, which are executed atomically, and possibly concurrently, in some order on the hardware. Each subprogram can consist of many elementary operations and can simultaneously use all subsystems. For example, for a matrix multiplication followed by bias addition and activation, the entire computation can be carried out in a single *fused* kernel (Snider & Liang, 2023; Google Cloud, 2024a), where matrix unit outputs are immediately processed by scalar/vector units (for bias addition and activation) before writing them to memory. This minimizes the overhead associated with launching and terminating kernels, avoids multiple round-trips to the memory, and enables more effective simultaneous use of the different compute resources. Figure 2 illustrates this concept.

### 2.3 Ridge point analysis

Depending on the kernel, different subsystems of an accelerator are utilized to varying degrees, with one often becoming the bottleneck that dictates the kernel's overall runtime. To accurately model performance and identify the bottleneck, we must quantify the capabilities of each accelerator subsystem and how the kernel uses it (Chern et al., 2022).

We quantitatively characterize an accelerator's performance by measuring the peak throughput of each of its subsystems. For example, we measure the peak memory bandwidth for each memory subsystem and the peak operations per second for each compute subsystem. Depending on the subsystems used and their extent of saturation, we may choose to model only the relevant subsystems. For example, if a subsystem is not used or contributes negligibly to the runtime, it can be omitted.

For ease of exposition, we present the analysis for TPUs and focus on three key subsystems: (i) HBM memory, (ii) the VPU and (iii) the MXU. However, this analysis applies to GPUs too, with MXU mapped to TensorCores and VPU to CUDA cores. We define the following parameters to quantify the throughput of each subsystem:

- $\beta$: maximum HBM bandwidth in bytes per second.

- $\gamma$: maximum number of VPU operations per second.

- $\pi$: maximum number of MXU operations per second.

Similarly, we characterize a kernel by measuring its utilization of each subsystem in its lifetime.

- $M$: number of bytes transferred to/from the HBM.

- $O_{\text{VPU}}$: number of operations executed on the VPU.

- $O_{\text{MXU}}$: number of operations executed on the MXU.

Since the kernel can utilize all subsystems simultaneously, its total runtime is determined by the subsystem that requires the most time to complete its work.[3] Therefore, we can estimate the total runtime of the kernel as:

$$\text{runtime} = \max\left(\frac{M}{\beta}, \frac{O_{\text{VPU}}}{\gamma}, \frac{O_{\text{MXU}}}{\pi}\right). \tag{1}$$

The bottleneck is the subsystem corresponding to the largest argument to $\max(\dots)$ in equation 1. For example, a kernel is considered to be *memory-bound* if its memory subsystem cannot feed data sufficiently fast to keep compute subsystems busy, i.e., $\frac{M}{\beta} \geq \max(\frac{O_{\text{VPU}}}{\gamma}, \frac{O_{\text{MXU}}}{\pi})$. To minimize overall runtime, we must address the bottleneck subsystem until it is no longer the limiting factor.

A corollary of this model is that increasing the utilization of non-bottleneck subsystems may *not* necessarily increase the kernel's runtime. To formalize this, we define *ridge points* of equation 1, which are configurations where the runtimes of any two subsystems are equal. For example, we can estimate as $\frac{\gamma}{\pi}$ the maximum number of VPU operations that can be performed per MXU operation to remain MXU-bound. Since the MXU has much higher throughput, i.e., $\pi \gg \gamma$, it is often more convenient to use the number of VPU operations per d-dimensional dot product on the MXU, i.e., $\frac{\gamma}{\left(\frac{\pi}{2d}\right)}$. This reformulation helps keep the ratio a small and interpretable integer. We can similarly define quantities such as $\frac{\gamma}{\left(\frac{\beta}{4}\right)}$ to denote the number of VPU operations that can be performed per 4-bytes of data transferred to/from the HBM. Ridge points thus provide a simple way to understand the balance between different subsystems and easily reason about how changes in a subsystem's utilization can impact overall performance (Williams et al., 2009). For a list of values of these quantities on different accelerators, see Table 1.

---

[3]In some cases, there may be dependencies between subsystems that may cause the dominating subsystem to stall. However, in practice, most optimized kernels do not suffer significantly from such dependencies and saturate at least one of the subsystems.

Table 1: **Peak throughput and ridge points of different subsystems in accelerators.** The $\gamma$ for TPUv4 was taken from Chern et al. (2022), and the $\gamma$ for TPUv5e was estimated by timing VPU-bound kernels (see Appendix A.1). All other values derived directly from the official datasheets.

| DEVICE | $\beta$ | $\gamma$ (TFLOP/s) | $\pi$ (TFLOP/s) | $\frac{\gamma}{\left(\frac{\pi}{256}\right)}$ | $\frac{\gamma}{\left(\frac{\beta}{4}\right)}$ |
|---|---|---|---|---|---|
| | | FP32 | BF16 | OPS PER 128-D DOT | OPS PER 4 BYTES |
| A100 PCIE | 1.935 TB/s | 19.5 | 312 | $\approx 16$ | $\approx 40$ |
| H100 SXM | 3.35 TB/s | 67 | 1,979 | $\approx 8$ | $\approx 80$ |
| TPUv4 | 1.2 TB/s | 4.3 | 275 | $\approx 4$ | $\approx 14$ |
| TPUv5E | 819 GB/s | $\approx 6.14$ | 197 | $\approx 8$ | $\approx 30$ |

## 3 Problem Setup

Given a matrix $W \in \mathbb{R}^{n \times d}$ and a vector $x \in \mathbb{R}^d$, the task is to approximately find the $K$ largest elements of $y := Wx \in \mathbb{R}^n$.

**Expected recall.** Let $U$ represent the set of actual top-K elements of $y$, and let $V$ represent the top-K elements returned by an approximate algorithm. We define *expected recall* as the expected fraction of the true top-K elements retrieved by the algorithm, assuming that the top-K elements are placed randomly and uniformly in $y$:

$$\mathbb{E}\left[\text{recall}\right] = \mathbb{E}\left[\frac{|U \cap V|}{|U|}\right].$$

**Objective.** The goal is to *minimize the time* required for this operation while *maximizing expected recall*, thereby improving the Pareto frontier describing the trade-off between latency and expected recall objectives.

## 4 Related Work

As of this writing, `jax.lax.top_k` is the only exact Top-K implementation available for TPUs. Chern et al. (2022) proposed a faster approximate Top-K algorithm, which is currently exposed in JAX as `jax.lax.approx_max_k`. To the best of our knowledge, these are the only Top-K implementations available for TPUs.

A large body of prior work focuses on exact Top-K algorithms for GPUs (Xie et al., 2024; Gaihre et al., 2021; Zhang et al., 2023; Li et al., 2024; Dashti et al., 2013; Alabi et al., 2012; Shanbhag et al., 2018; Ribizel & Anzt, 2019). Since the algorithmic framework introduced by Chern et al. (2022), which we generalize in this work, can transform any exact algorithm into a faster approximate variant, we do not compare with exact algorithms. We provide a detailed description of the Chern et al. (2022) algorithm in Section 5.

Key et al. independently develop the same generalized algorithm, implement it for GPUs in the unfused setting, and evaluate quality in several downstream tasks including SparQ attention Ribar et al. (2024) and link-prediction in knowledge graphs. Although the algorithm is shared, our work provides several distinct theoretical and practical advances. Our probabilistic model of the algorithm is exact, whereas theirs relies on approximations that lead to noticeable differences (see Appendix Section A.6). They provide an API, `approx_top_k(array, K, K', B)`, that requires manual tuning of algorithm parameters. In contrast, with our theoretical analysis along with detailed runtime performance modeling, we provide a fast automatic parameter selection routine and a more user-friendly interface, `approx_top_k(array, K, recall_target)`. Beyond the unfused setting, we also implement a matmul-fused implementation that extracts non-trivial speedups in realistic workloads. The ridge-point analysis in Section 2.3, combined with our analysis of the implementation in Section 6.3, provides a principled framework for determining the available headroom in a fusion and enables fusion-aware parameter selection, which could, in principle, be adapted into a cost model to support automatic compiler-driven fusion decisions.

# 5    The Original Algorithm

Chern et al. (2022) designed an approximate Top-$K$ algorithm that operates in two stages. In the first stage, the input array is partitioned by grouping elements separated by a fixed stride into buckets. The top-1 element of each bucket is gathered to form a *smaller* array. In the second stage, this array is sorted using bitonic sort, and the top K elements are returned. The first stage reduces the size of the input array for the *expensive* second stage, which improves performance. Approximations occur when multiple top-K elements fall into the same bucket since only one is selected and the rest are discarded. Increasing the number of buckets reduces the likelihood of such collisions and can be chosen to achieve a desired average recall target. Appendix Figure 5 presents the algorithm.

The algorithm accepts `recall_target` as a parameter, denoted by $r$, which specifies the desired expected recall. The required number of buckets, denoted by $B$, is calculated using a closed-form expression that relates $B$ to the expected recall under a model in which the positions of the top-K elements are independently and uniformly distributed over the input array:

$$B \geq \frac{1}{1 - \mathbb{E}[\text{Recall}]^{\frac{1}{K-1}}} \approx \frac{K-1}{1-r}.$$

The input often results from a matrix multiplication, e.g., a maximum inner-product search (MIPS) or the Top-$K$ on key-query logits in attention. The algorithm's first stage, which executes on the scalar/vector units, can often be fused with the preceding matrix multiplication, which executes on the matrix units. Hence, the fused first stage might incur little to no additional cost since it utilizes the otherwise idle scalar/vector compute units while the matrix units are busy.

To design their algorithm and fused implementation, the authors adopt a principled approach by modeling accelerator performance, similar to the model described in Section 2.3. Their first stage uses a fixed budget of three operations per element to track the top-1 element (and its index) of each bucket. We argue that this leaves compute resources underutilized in many cases, as detailed below:

1. Their analysis focuses on matrix multiplications with 128-dimensional dot products, which on most hardware provide only 4-8 vector operations per output element, i.e., each 128-d dot product, consistent with their budget of three operations per element. However, we frequently work with larger dimensions, where the available scalar/vector compute per output element is nearly $\frac{\text{dims}}{128}$ times higher than the numbers they estimate.

2. Even for 128-dimensional dot products, the first stage may not saturate the scalar/vector units on all hardware platforms. See Table 1.

3. In memory-bound computations, more scalar/vector compute is available than would be expected from a matrix-multiplication-bound computation.

The additional compute available enables more sophisticated algorithms for the first stage, potentially improving the recall with fewer elements to process in the second stage. A more expensive first stage might still yield overall gains if gains in the second stage outweigh the increased cost of the first stage. Based on these insights, we generalize their algorithm to more *flexibly* utilize the available compute by selecting *the top-$K'$ elements from each bucket instead of just the top-1*.

# 6    Method

We describe our algorithm in Section 6.1 and provide an analysis in Section 6.2. In Section 6.3, we discuss the key ideas in our implementation of the algorithm for TPUs.

### 6.1 Algorithm

**Step 1: Partition into buckets**

Given $A = [a_1, \ldots, a_N]$, partition into $B$ buckets:

$$G_i = \{a_{i+jB} \mid j \in \mathbb{Z}_{\geq 0}, i + jB \leq N\}, \quad i = 1, \ldots, B$$

**Step 2: Select Top-$K'$ per bucket**

Select the top-$K'$ elements from each bucket:

$$T_{K'}(G_i) = \text{Top-K'}(G_i)$$

**Step 3: Merge selections**

Merge the top-$K'$ elements from all buckets:

$$A_{\text{selected}} = \bigcup_{i=1}^{B} T_{K'}(G_i)$$

**Step 4: Return approximate top-$K$**

Sort the merged set and return the top-$K$ elements:

$$T_K^{\text{approx}}(A) = \text{sorted}(A_{\text{selected}})[: K]$$

### 6.2 Analysis

Consider a scenario in which we have $N$ balls, $K$ of which are special balls, and $B$ buckets. The $N$ balls are evenly distributed in the $B$ buckets. To model the distribution process, we can randomly order all the balls and then partition them into buckets: the first $\frac{N}{B}$ balls go to the first bucket, the next $\frac{N}{B}$ balls go to the second bucket, and so on. In the context of our algorithm, the $N$ balls correspond to the input elements, the $K$ special balls represent the top-K elements, and $B$ buckets correspond to the "buckets" in the algorithm.

Let $X_b$ be a random variable that denotes the number of special balls in bucket $b$. Approximation errors occur when more than $K'$ special balls are placed in the same bucket. The total number of excess collisions is given by the sum of excess special balls in each bucket.

$$\mathbb{E}\left[\text{Excess-collisions}\right] = \mathbb{E}\left[\sum_{b=1}^{B} \max(0, X_b - K')\right]$$
$$= \sum_{b=1}^{B} \mathbb{E}\left[\max(0, X_b - K')\right].$$

There exists a joint probability distribution that governs the set of $X_b$ that satisfies the constraint that the total number of special balls in all buckets sums to $K$, i.e., $\sum_{b=1}^{B} X_b = K$. However, the marginals $X_b$ are all identically distributed as $X_b \sim \text{Hypergeometric}(N, K, \frac{N}{B})$. This arises because the distribution of special balls in the first bucket must be the same as in all other buckets by symmetry, and it is apparent that the distribution of special balls in the first bucket must follow $\text{Hypergeometric}(N, K, \frac{N}{B})$. This is sufficient to simplify further:

$$\mathbb{E}\left[\text{Excess-collisions}\right] = B \times \mathbb{E}\left[\max(0, X_0 - K')\right].$$

The number of excess collisions is related to the recall as follows.

$$\mathbb{E}\left[\text{Recall}\right] = 1 - \frac{\mathbb{E}[\text{Excess-collisions}]}{K}.$$

In Appendix A.3, we verify the accuracy of Monte Carlo evaluations of this expectation against the recall obtained from the simulated runs of the algorithm. Theorem 1 derives an algebraic expression for this expectation.

Chern et al. (2022) model their algorithm as randomly distributing $K$ balls in $B$ buckets and relate it to the classical birthday problem. Based on this model, they derive a bound on the expected recall and invert the expression to obtain a formula for the number of buckets. However, their analysis neglects several structural constraints of the problem: (i) the number of balls in each bucket cannot exceed $\frac{N}{B}$, (ii) the balls are sampled without replacement, and (iii) only the non-colliding balls are counted as correctly retrieved, even though *one* of the colliding balls is always correctly retrieved in a bucket with collisions. *In Theorem 1, we derive a new bound on the number of buckets for $K' = 1$ based on our model that is provably tighter than theirs by at least a factor of two.* We verify the quality of our bounds in Appendix A.5 and show that it closely approximates the exact expression with high fidelity.

**Theorem 1.** *Suppose $N$ balls are randomly distributed into $B$ buckets $G_1, \cdots, G_B$, each getting $N/B$ balls. The recall of the top-$K'$ balls across all the $B$ buckets with respect to the top-$K$ balls overall is given by:*
$$\mathbb{E}[Recall] = 1 - \frac{B}{K} \times \sum_{r=K'+1}^{\min(K,N/B)} (r - K') \frac{\binom{K}{r}\binom{N-K}{\frac{N}{B}-r}}{\binom{N}{\frac{N}{B}}}.$$ *Specifically, for $K' = 1$ and a target recall factor of at least $r$, the bound below implies that $B = \frac{K}{2\left(1 - r + \frac{K}{2N}\right)}$ suffices to guarantee the target recall $r$.*

**Remark**. Note that for $K' = 1$, the bound is a factor of 2 tighter than that in Chern et al. (2022).

We provide the proof in Appendix A.4.

### 6.3 Implementation

In the first stage, we take an input array of shape `[batch_size, N]` and output two vectors: one for values and another for indices, both of which have the shape `[batch_size, B × K']`. Here, $N$ is the total number of elements, $B$ is the number of buckets, and $K'$ is the number of top elements we select from each bucket.

We focus on identifying the top-$K'$ elements of a single bucket since supporting multiple buckets is a matter of running many *independent* instances of this subroutine. To create an effective fusible implementation, we track the top-$K'$ elements in an online fashion as inputs continuously stream in from the matrix multiplication unit. We maintain two lists per bucket, one for the top-K' `values` and another for their corresponding `indices`. The `values` list is kept in descending order, and we ensure that each value's corresponding index is at the same position in the `indices` list. When a new element arrives, we update the lists in two steps:

1. If the new element is larger than the smallest element in the `values` list, we replace the smallest element (and its index) with the new element (and its index).

2. We perform a single bubble sort pass over the lists to move the new element to its correct position.

Algorithm 1 contains the pseudocode for this subroutine. The first step requires one comparison and two selects for updating the value and index. The second step requires comparing adjacent elements (one comparison) and conditionally swapping elements (four selects) for each of the $(K' - 1)$ positions. In total, each input element requires $(5K' - 2)$ operations.

Since the `values` list is stored in descending order, an input element larger than the k-th value in the list is also larger than all subsequent values. This property allows the comparison in Line 9, Algorithm 1 to be done using the input element as the LHS, which eliminates a loop-carried dependency.

Note that Algorithm 1 does not include an early return if the condition on Line 4 fails, nor does it exit the bubble sort loop early when the condition in Line 9 fails. This is required to vectorize the routine across buckets. An early return would theoretically reduce operations for individual buckets but would prevent vectorization across buckets.

Once all inputs are processed, we obtain the final result by separately merging all `values` lists to obtain the first-stage values list and merging the corresponding `indices` lists to obtain the first-stage indices list.

Since buckets group elements separated by a fixed stride, contiguous input elements map to different buckets. We can logically view the input array as having the shape `[batch_size, N / B, B]`. We store the top-K' lists with a physical layout of `[batch_size, K', B]` so that the minor-most axis maps to the bucket axis,

| **Algorithm 1** `TopKPrimeUpdate` | **Algorithm 2** Vectorized Top-$K'$ Update |
|---|---|
| 1: **Input:** input, index, values[K'], indices[K'] | 1: **Input:** input array of size $N$, number of lanes $L$, number of buckets $B$, top-$K'$ values (-$\infty$ initialized) and indices lists of shape [B, K'] |
| 2: **Output:** values[K'], indices[K'] | 2: **Output:** top-$K'$ values and indices lists |
| 3: **Precondition:** values sorted in descending order | 3: $num\_chunks \leftarrow N/L$ |
| 4: **if** input $\geq$ values[K'] **then**  /* one compare */ | 4: **for** $in\_chunk \leftarrow 0$ **to** $num\_chunks - 1$ **do** |
| 5:    values[K'] = input  /* one select */ | 5:    $out\_chunk \leftarrow in\_chunk \bmod (B/L)$ |
| 6:    indices[K'] = index  /* one select */ | 6:    **Load** inputs corresponding to $in\_chunk$ |
| 7: **end if** | 7:    **Load** current top-$K'$ lists for $out\_chunk$ |
| 8: **for** $k = K'$ **to** 2 **do** | 8:    **Compute** updated top-$K'$ lists using trivially vectorized `TopKPrimeUpdate` |
| 9:    **if** values[k] > values[k-1] **then** /* one compare */ | |
| 10:      swap(values[k], values[k-1])  /* two selects */ | 9:    **Store** updated top-$K'$ list at $out\_chunk$ |
| 11:      swap(indices[k], indices[k-1]) /* two selects */ | 10: **end for** |
| 12:    **end if** | |
| 13: **end for** | |

which aligns with the input's logical shape. The top-K' update subroutine (Algorithm 1) can be executed independently for each bucket and is trivially vectorizable along the bucket axis.

To simplify the implementation, we restrict the number of buckets to a multiple of the vector width, denoted by $L$. We process contiguous L-sized chunks of the inputs and their corresponding L-sized top-K' lists in each iteration of a vectorized loop. Algorithm 2 outlines the vectorized version. Although this implementation appears to require $2K'$ loads and stores of the top-K' lists for each input that is read, we can schedule the loop iterations so that the input chunks corresponding to the same bucket are executed consecutively; this lets the top-K' lists fully reside in the registers or the nearest cache depending on the choice of $K'$ and the hardware.

Based on these insights, we implement our first-stage kernel in Pallas, a JAX kernel language (Bradbury et al., 2018). We use `jax.lax.sort_key_val` and slice the top-K elements for the second stage. We share the Python code for our algorithm with detailed comments in Listing A.8 for the unfused implementation and Listing A.9 for the matmul-fused implementation. To find the algorithm parameters for a given input shape and recall target, we sweep through legal configurations and list those that meet the recall target. We then heuristically choose the configuration with the best performance. To calculate expected recall, we use Monte Carlo evaluations of the expectation expression derived in Section 6.2. We share the Python code to estimate expected recall and select algorithm parameters in Listing A.10. The computational cost of this procedure is relatively low; see Appendix Section A.10.2 for a detailed discussion.

## 7    Results and Discussion

In Section 7.1, we show that our algorithm substantially reduces the input size for the same expected recall compared to the improved baseline, which is the original algorithm by Chern et al. (2022) with our improved bound. Section 7.2 discusses the performance of our unfused Pallas implementation for TPUs, and Section 7.3 does the same for our matmul-fused implementation.

### 7.1    Theoretical effectiveness of the first-stage filtering

Table 2 shows the relationship between $K'$, the number of buckets, and the expected recall to select the top-1024 elements of an array of 262,144 elements. With a fixed number of output elements ($K' \times num\_buckets$), the expected recall increases significantly with $K'$. Keeping the expected recall fixed, even small values of $K'$ ($\leq 4$) substantially reduce the number of output elements. For example, to achieve an expected recall of 95%, $K' = 1$ requires at least 16,384 output elements, while $K' = 4$ requires only 2,048 elements, an 8$\times$ reduction in the number of elements to process.

Table 2: **(Left) Expected recall versus K' for selecting top-1024 elements from an array of 262,144 elements.** `#num_elements` refers to the number of output elements from the first stage, which is $B \times K'$. A smaller `#num_elements` leads to better performance in the second stage. **(Right) The runtime of our algorithm on TPUv5e for a batch size of 8.** The `jax.lax.approx_max_k` rows present the performance of the official JAX implementation (which supports only $K' = 1$), while the $K' = 1$ rows present the performance of our implementation.

| ALGORITHM PARAMETERS | | ALGORITHMIC PERFORMANCE | | RUNTIME PERFORMANCE | | |
| K' | BUCKETS | NUM_ELEMENTS | $\mathbb{E}$[RECALL] | STAGE 1 | STAGE 2 | TOTAL |
|---|---|---|---|---|---|---|
| JAX.LAX.APPROX_MAX_K | 131,072 | 131,072 | $0.998 \pm 0.000$ | 12US | 649US | 661US |
| JAX.LAX.APPROX_MAX_K | 65,536 | 65,536 | $0.992 \pm 0.001$ | 13US | 292US | 305US |
| JAX.LAX.APPROX_MAX_K | 32,768 | 32,768 | $0.987 \pm 0.004$ | 13US | 131US | 144US |
| 1 | 65,536 | 65,536 | $0.992 \pm 0.001$ | 13US | 313US | 326US |
| 1 | 32,768 | 32,768 | $0.987 \pm 0.004$ | 14US | 141US | 155US |
| 1 | 16,384 | 16,384 | $0.972 \pm 0.005$ | 13US | 64US | 77US |
| 1 | 8,192 | 8,192 | $0.942 \pm 0.007$ | 13US | 30US | 42US |
| 2 | 4,096 | 8,192 | $0.991 \pm 0.003$ | 15US | 30US | 45US |
| 2 | 2,048 | 4,096 | $0.968 \pm 0.006$ | 13US | 14US | 27US |
| 3 | 2,048 | 6,144 | $0.996 \pm 0.002$ | 16US | 32US | 48US |
| 3 | 1,024 | 3,072 | $0.977 \pm 0.005$ | 12US | 11US | 23US |
| 4 | 1,024 | 4,096 | $0.996 \pm 0.002$ | 13US | 14US | 27US |
| 4 | 512 | 2,048 | $0.963 \pm 0.007$ | 12US | 8US | 20US |
| 5 | 512 | 2,560 | $0.989 \pm 0.004$ | 13US | 9US | 22US |
| 6 | 512 | 3,072 | $0.997 \pm 0.002$ | 14US | 11US | 25US |
| 6 | 256 | 1,536 | $0.951 \pm 0.008$ | 14US | 8US | 22US |
| 8 | 512 | 4,096 | $0.992 \pm 0.004$ | 16US | 14US | 30US |
| 10 | 256 | 2,560 | $0.999 \pm 0.000$ | 19US | 9US | 28US |
| 12 | 128 | 1,536 | $0.984 \pm 0.006$ | 23US | 8US | 31US |
| 16 | 128 | 2,048 | $0.999 \pm 0.001$ | 29US | 8US | 37US |

Appendix Figure 10 plots expected recall versus the number of output elements for different values of $K'$ to select the top-3,360 elements from an array of 430,080 elements. The expected recall improves rapidly with increasing $K'$, as highlighted by the clear separation between the curves corresponding to our algorithm and the baseline.

Figure 3 shows the factor by which our algorithm variants, up to $K' = 4$, reduce the size of the second-stage input across a broad range of configurations ($\frac{K}{N} \in \{0.01\%, \ldots, 25\%\}$ and array sizes $\in [256, 4e9]$) over the baseline $K' = 1$. We account for the implementation constraints necessary for simplicity and performance such as requiring the number of buckets to be a multiple of 128 and divide the input array size evenly; therefore, the numbers indicate *real realizable reductions* using our implementation. The figure demonstrates that our algorithm significantly reduces the number of elements in virtually all configurations, with a median reduction of 7×. In particular, the $K' > 1$ variants perform worse or equal to $K' = 1$ only for very small values of K ($K \leq 10$), which is a consequence of our implementation constraints where a smaller $K'$ can sometimes yield a better-aligned $B$ that results in fewer total $B \cdot K'$ elements after rounding. However, since we always select the best $K'$ in $[1, 4]$, our method never performs worse than the baseline by construction. We conclude that our algorithm is broadly applicable and effectively reduces the number of output elements.

## 7.2 Improved latency of finding the Top-K elements on TPUv5e

Table 2 presents the latency of our unfused implementation to identify the top-1024 elements from an array of 262,144 elements. To achieve a recall target of 99%, the baseline requires 305µs, while our algorithm with $K' = 4$ takes only 27µs, resulting in an 11× speedup.

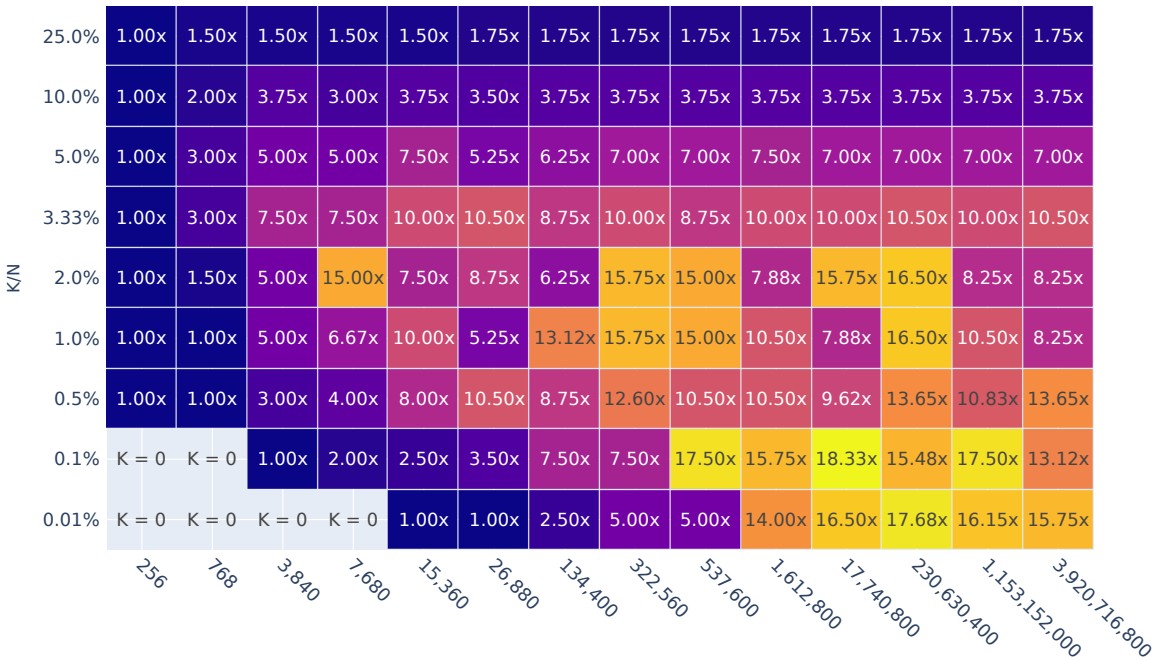

Figure 3: **Factor of reduction in output elements over the baseline ($K' = 1$) for 99% expected recall target.** The heatmap shows the factor by which our generalized algorithm with $1 \leq K' \leq 4$ reduces the number of elements in the first stage over the reductions provided by the baseline, i.e., a value of $2\times$ indicates that our algorithm outputs two times fewer elements compared to the $K' = 1$ baseline. Our implementation constrains the number of buckets to be a multiple of 128 and divide the input size for simplicity and performance, which is accounted for in this figure. Even though $K' > 1$ would require fewer buckets compared to $K' = 1$, rounding the number of buckets to satisfy the constraints may lead to more output elements than required by $K' = 1$.

The first stage cost remains nearly constant from $K' = 1$ to $K' = 6$ due to its memory-bound nature. According to the performance model in Section 2.3 and Table 1, the first stage must be memory bound until we exceed 30 VPU operations per 4-byte element, which occurs around $K' = 6$, according to the operation count formula in Section 6.3. Therefore, we expect the latency of the first stage to be independent of $K'$ until we reach $K' = 6$.

## 7.3 Fusing Top-$K$ with matrix multiplication

Many real-world applications require identifying the Top-$K$ results from the outputs of matrix multiplications. One prominent example is maximum inner-product search (MIPS), where, for a given query vector, the task is to retrieve the top-K vectors from a large database that have the highest inner products.

We evaluate our algorithm on a MIPS workload of one million 128-dimensional vectors and 1024 queries. Table 3 reports runtimes on TPUv5e. Exact Top-$K$ (`jax.lax.top_k`) takes $80\times$ longer in the second stage than the matmul (587ms vs 7.32ms). `jax.lax.approx_max_k` reduces this to $13\times$ (118ms), and our $K' = 1$ unfused implementation to $6\times$ (50ms).

Moving to $K' = 4$, the second stage (3.51ms) falls below half the matmul cost, leaving the matmul (7.31ms) and first stage (10.80ms) as the bottleneck. Fusing the first stage with the matmul eliminates its cost and improves matmul performance (6.55ms). The gains from fusion can be significant in practice. The

Table 3: **The runtime of our algorithm on TPUv5e to identify the top-1024 elements from a database of 1M 128-dimensional vectors with 99% recall for 1024 query vectors.** The `jax.lax.top_k` row represents the performance of exact Top-K. The `jax.lax.approx_max_k` row presents the performance of the official JAX implementation for the $K' = 1$ setting. The remaining rows present the performance of our implementation.

| ALGORITHM | MATMUL | STAGE 1 | STAGE 2 | TOTAL |
|---|---|---|---|---|
| JAX.LAX.TOP_K | 7.32MS | – | 587MS | 594MS |
| JAX.LAX.APPROX_MAX_K | 9.06MS | FUSED | 118MS | 127MS |
| $K' = 1$ | 7.32MS | 6.58MS | 50.0MS | 64MS |
| $K' = 1$ | 9.03MS | FUSED | 50.0MS | 59MS |
| $K' = 4$ | 7.31MS | 10.80MS | 3.51MS | 22MS |
| $K' = 4$ | 6.55MS | FUSED | 3.51MS | 10MS |

MIPS matmul multiplies `[B, D]` by `[D, N]` (queries, vector size, database size), giving arithmetic intensity $\frac{2}{E} \min(B, D)$ (derived in Appendix A.12), where $E$ is the element size. In large-scale deployments with $D$ in the low hundreds, the matmul is often memory-bound. Fusion avoids writing the large output tensor to memory, increasing arithmetic intensity and shifting the matmul closer to compute-bound.

## 8 Conclusion

We present a generalization of the approximate Top-$K$ algorithm of Chern et al. (2022) that selects the top-$K'$ elements per bucket in the first stage, instead of restricting to top-1. We motivate this generalization through a principled analysis of the runtime behavior of the original algorithm, showing that the choice of $K' = 1$ does not fully utilize the available compute resources. This observation naturally leads to a broader family of algorithms parameterized by $K'$, which can substantially reduce the second-stage workload while maintaining high recall. We theoretically analyze the generalized algorithm, deriving an exact expression for the expected recall, and obtain a bound for $K' = 1$ which is provably tighter by a factor of 2 compared to that in Chern et al. (2022). For $K' > 1$, we demonstrate both theoretically and empirically that the algorithm substantially reduces the second-stage input size across a wide range of configurations, spanning $\frac{K}{N}$ ratios from 0.01% to 25%, with a median reduction of 7× over the $K' = 1$ baseline.

To realize these gains in practice, we implement the algorithm for Cloud TPUv5e together with a principled performance modeling framework that accurately predicts runtimes and enables a user-friendly `approx_top_k(array, K, recall_target)` interface that requires no manual tuning. We additionally present a matmul-fused implementation that further eliminates the first-stage cost. While our kernel implementations are specific to TPUs, the algorithm, performance modeling and its analysis are general, and we believe that it has applicability to other hardware platforms such as GPUs.

**Limitations and Future Work.** Our empirical evaluation is focused on Cloud TPUv5e, and extending empirical evaluation and the matmul-fused implementation to GPUs is an important open direction. The recall analysis assumes uniform random placement of top-$K$ elements; efficient parallel randomization schemes that bring real-world inputs in line with this model would fully close the gap between theory and practice. While we derive a provably tighter bound for the $K' = 1$ setting, extending similarly tight numerically stable analytical bounds to the general $K' > 1$ setting remains open. In particular, deriving closed-form approximations that eliminate the need for Monte Carlo estimation during parameter selection would improve the practicality of the approach. A further limitation is that our analysis focuses on expected recall and does not characterize its variance or the full error distribution. In particular, understanding the variability induced by collisions could yield a more complete picture of the algorithm. Integrating the algorithm into an optimizing compiler would enable automatic fusion decisions and fusion-aware parameter selection. Our generalization of the first stage represents the simplest step in a richer design space; more sophisticated first-stage selection schemes beyond the top-$K'$ per bucket approach could push the performance further.

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

## A  Appendix

### A.1  Estimating peak VPU throughput of TPUv5e.

We used two test programs with a controllable parameter that allows us to vary the number of vector operations per element. We run the programs on a large `s32[4096, 4096]`-shaped array with different parameters and time the kernels. We verify that the compiler fuses the operations into a single kernel. We assume that addition and multiplication are instructions in the TPU's ISA. Given the large size of the inputs, we assume that these programs saturate the vector processing unit.

```
1  @partial(jax.jit, static_argnames='n')
2  def fibonacci(x, y, n):
3    for i in range(n):
4      c = x + y
5      x = y
6      y = c
7    # We expect the compiler to optimize the snippet to the following
         sequence:
8    # r1 = x
9    # r2 = y
10   # r3 = r1 + r2
11   # r1 = r2 + r3
12   # r2 = r3 + r1
13   # r3 = r1 + r2
14   # ...
15   #
16   # For every two elements read from memory, we perform 'n' additions.
17   return y
```

```
1  @functools.partial(jax.jit, static_argnames='steps')
2  def fast_exponentiation(x: jax.Array, steps: int):
3    z = x
4    for _ in range(1, steps):
5      z = z * z
6    return z
```

Figure 4: **Estimating the throughput of the VPU on TPUv5e.** We expect the kernels to be memory-bound (constant line) initially and then be vector compute bound (linear scaling). We fit a line to the points in the linear region with the following model: $time = num\_ops \times \frac{1}{throughput} + overhead$. The inverse of the slope is an estimate of the peak throughput of the VPU.

## A.2   Visual illustration of the Chern et al. (2022) algorithm.

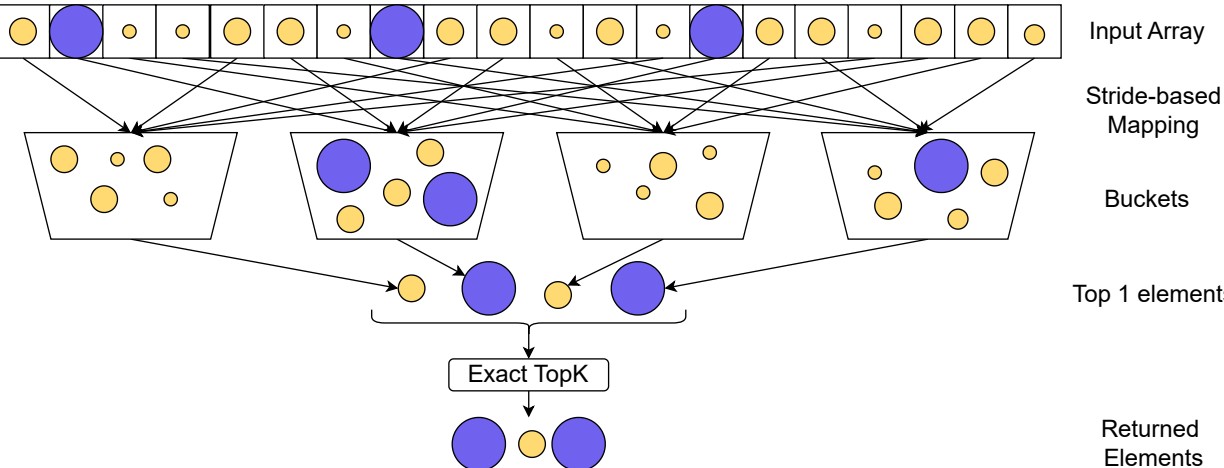

Figure 5: **The two-stage approximate Top-$K$ algorithm by (Chern et al., 2022).** This example demonstrates the process of finding the approximate top three elements from an array of twenty elements using the algorithm by Chern et al. (2022). Ten buckets are required to guarantee an expected recall of 85%, but we use only four for illustration purposes. The size of the balls indicates their value, and the top three balls have been colored blue for visual clarity. The first stage groups elements separated by a fixed stride of four into buckets and selects the top-1 element from each bucket. An exact Top-$K$ algorithm is applied on the selected elements to obtain the final result. In this example, two of the three actual top balls map to the same bucket, and one is dropped, resulting in an approximation error.

### A.3 Empirical verification of the quality of Monte Carlo estimates for expected recall.

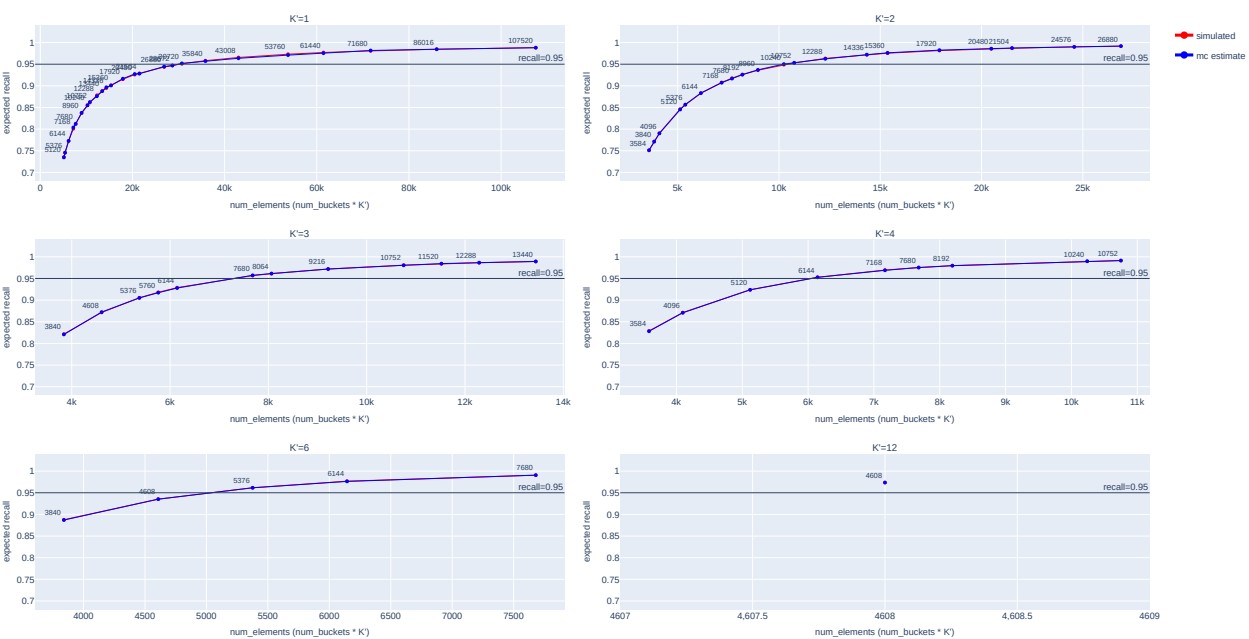

Figure 6: **Verification of Monte Carlo estimates of expected recall against simulated runs of the algorithm for finding top-3360 ($\approx 0.8\%$) elements from an array of size 430,080.** The simulated estimates were obtained from 1024 runs of the algorithm on randomly generated integers and the Monte Carlo estimates were obtained from 262144 samples of the expectation expression derived in Section 6.2.

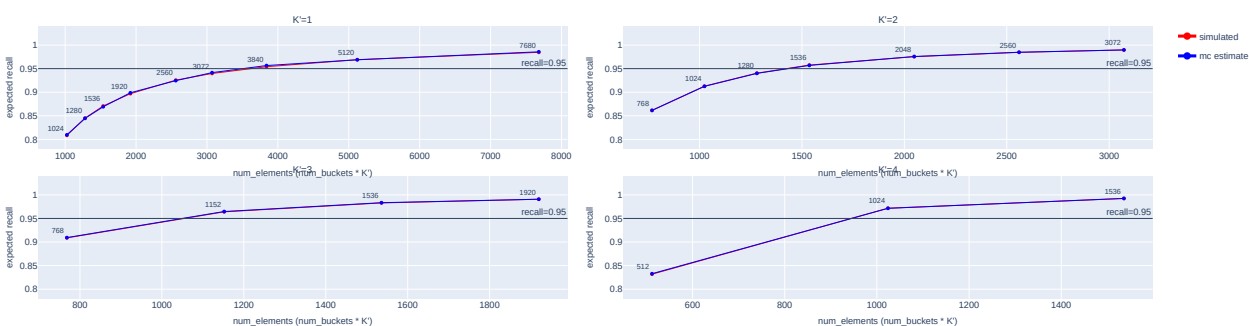

Figure 7: **Verification of Monte Carlo estimates of expected recall against simulated runs of the algorithm for finding top-480 ($\approx 3\%$) elements from an array of size 15,360.** The simulated estimates were obtained from 1024 runs of the algorithm on randomly generated integers and the Monte Carlo estimates were obtained from 262,144 samples of the expectation expression derived in Section 6.2.

### A.4 Proof of Theorem 1.

*Proof.* (Proof of Theorem 1) Consider an arbitrary subset $S \subseteq \{1, \cdots, N\}$ such that $|S| = K$. Let $S_j := S \cap G_j$ for $j = 1, \cdots, B$. We now want to bound the number of elements in $S_j$ greater than $K'$ for some given $K'$:

$$m_j := \mathbb{E}\left[\max\left(0, |S_j| - K'\right)\right]$$
$$= \sum_{r=K'+1}^{\min(K, N/B)} (r - K') \frac{\binom{K}{r}\binom{N-K}{\frac{N}{B}-r}}{\binom{N}{\frac{N}{B}}},$$

where each term in the summation refers to having $|S_j| = r$; $\binom{K}{r}$ refers to choosing $r$ elements out of $S$; $\binom{N-K}{\frac{N}{B}-r}$ refers to the number of subsets, where $\frac{N}{B} - r$ elements in $G_j$ are chosen from outside of $S$; and $\binom{N}{\frac{N}{B}}$ refers to the total number of possible subsets that $G_j$ can take. Finally, the recall (i.e., the expected number of elements in $S$ eventually captured by the output of our algorithm) is given by:

$$\mathbb{E}\left[\text{Recall}\right] = 1 - \frac{B \cdot m_j}{K}.$$

We now show that the above expression is provably tighter than the expression obtained in Chern et al. (2022) for $K' = 1$. Specifically, for $K' = 1$, note that:

$$\mathbb{E}\left[|S_j| - K'\right] = -1 \cdot \mathbb{P}\left[|S_j| = 0\right] + m_j \tag{2}$$
$$\Rightarrow m_j = \mathbb{E}\left[|S_j|\right] - 1 + \mathbb{P}\left[|S_j| = 0\right] \tag{3}$$
$$\Rightarrow m_j = \frac{K}{B} - 1 + \frac{\binom{N-K}{\frac{N}{B}}}{\binom{N}{\frac{N}{B}}} \tag{4}$$
$$\Rightarrow m_j \leq \frac{K}{B} - 1 + \left(1 - \frac{K}{N}\right)^{\frac{N}{B}} \tag{5}$$
$$\Rightarrow m_j \leq \frac{K}{B} - 1 + 1 - \frac{N}{B}\frac{K}{N} + \binom{\frac{N}{B}}{2}\left(\frac{K}{N}\right)^2 \tag{6}$$
$$\Rightarrow m_j \leq \frac{K^2}{2B}\left(\frac{1}{B} - \frac{1}{N}\right). \tag{7}$$

From the above, we see that the expected recall can be bounded as:

$$\mathbb{E}\left[\text{Recall}\right] \geq 1 - \frac{B}{K} \cdot \frac{K^2}{2B}\left(\frac{1}{B} - \frac{1}{N}\right)$$
$$= 1 - \frac{K}{2}\left(\frac{1}{B} - \frac{1}{N}\right),$$

or equivalently, if we choose

$$B = \frac{K}{2\left(1 - r + \frac{K}{2N}\right)},$$

we will then have $\mathbb{E}\left[\text{Recall}\right] \geq r$. On the other hand, Chern et al. (2022) use $B = \frac{K}{1-r}$ to guarantee a recall of $r$, which is more than twice as large as required by our formula.

$$\underbrace{\frac{1}{2} \cdot \frac{K}{\left(1 - r + \frac{K}{2N}\right)}}_{\text{our formula}} < \frac{1}{2} \cdot \left(\frac{K}{1 - r}\right) < \underbrace{\frac{K}{1 - r}}_{\text{their formula}}$$

In Appendix Section A.5, we verify the tightness of our bound and show that expanding up to the quartic term in step 6 provides a near-perfect approximation of the exact expression that is practically indistinguishable. $\square$

## A.5 Quality of the theoretical bounds on expected recall for $K' = 1$ setting.

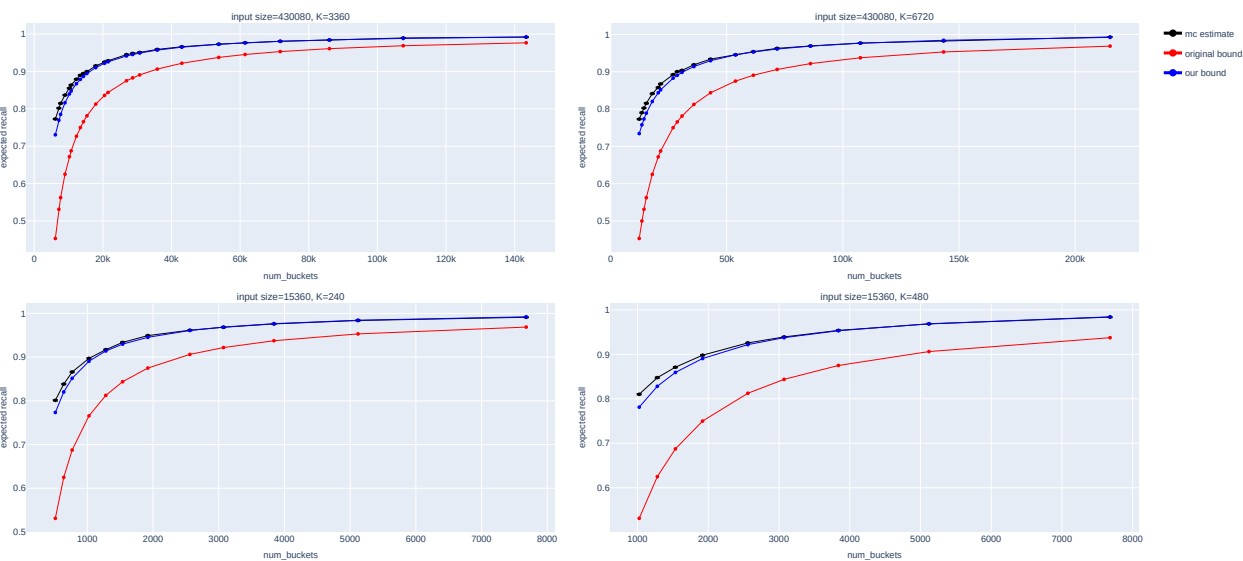

Figure 8: **Tightness of our theoretical bound on expected recall for $K' = 1$ setting compared to the original bound derived in Chern et al. (2022)**. See Section 6.2 for the derivation of our bound ($\mathbb{E}[\text{Recall}] \geq 1 - \frac{K}{2B}$) which is tighter than the original bound ($\mathbb{E}[\text{Recall}] \geq 1 - \frac{K}{B}$).

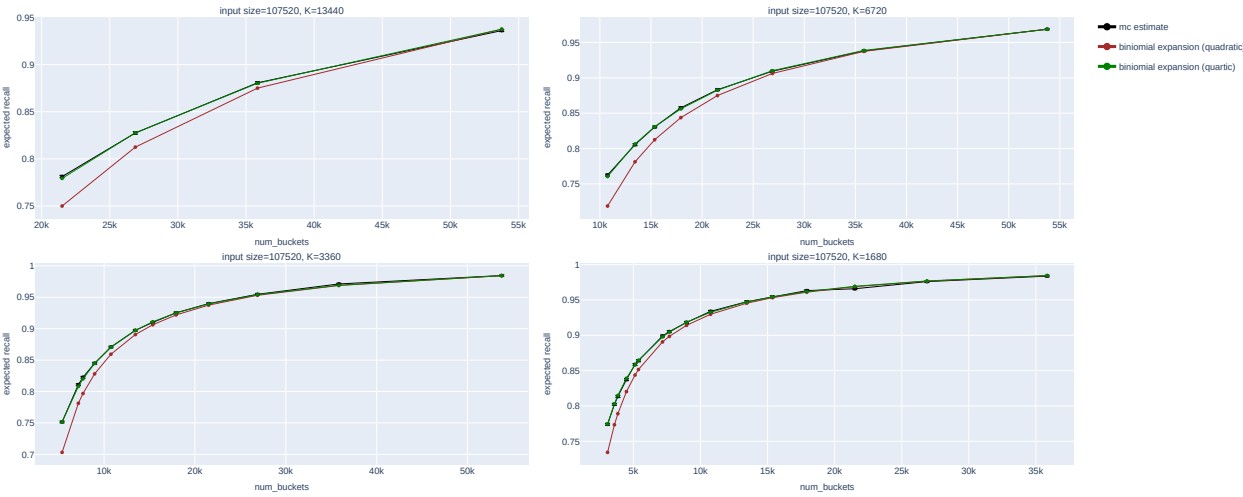

Figure 9: **Expanding the binomial expression in Step 6 to quartic terms in Theorem 1 approximates the expected recall that is nearly exact.**

### A.6 Comparison to the work by Key et al. (2024).

We distinguish our theoretical analysis from the work of Key et al. (2024) primarily through the rigor and conceptual simplicity of our model. The recall expression in Key et al. (2024) (Appendix C) is approximate rather than exact. Specifically, because the algorithm distributes the $N$ elements into $B$ buckets, the distribution of the actual Top-$K$ elements in each bucket follows the hypergeometric distribution, not the binomial distribution. Therefore, their resulting expression is independent of $N$, which is only an asymptotic approximation. As shown in our Appendix Section A.5, the dependence on $N$ is non-negligible in low recall regimes; our exact derivation (Appendix Section A.3) captures this variance, leading to essentially tight bounds for the $K' = 1$ case (Appendix Section A.4). Furthermore, our derivation is conceptually simpler and avoids state tracking or recursive expressions used in Key et al. (2024).

### A.7 Packages and utility functions for the code listings.

```python
import jax
import jax.numpy as jnp
from jax.experimental import pallas as pl

import numpy as np

def get_all_factors(n):
    small_factors = [i for i in range(1, int(np.ceil(np.sqrt(n)))) if n % i
        == 0]
    pair_factors = [n // factor for factor in small_factors]
    return set(small_factors + pair_factors)
```

### A.8 Unfused implementation of our algorithm

```python
def generalized_partial_reduce(inputs, local_K, num_buckets,
    tunable_params={}, **kwargs):
    """ApproxTopK with generalized partial reduce for minor-axis reductions.

    Note: Input elements separated by a fixed stride form a bucket.

    Args:
        inputs: jax.ShapeDtypeStruct-compatible object of the input array of
            the
            shape [batch_size, reduction_dims].
        local_K: number of top elements to keep track of per bucket
        num_buckets: number of buckets
        tunable_params: These are hardware-specific (auto)tunable parameters.
            See
            the uses in the function body for more information.
        *kwargs: forwarded as is to `pallas_call`.

    Note: The default tunable params selection does not check if the choices
        are
        meaningful (e.g., VMEM OOMs). Please tune the choices if they don't
            work.

    Note: The procedure does not allow arbitrary input shapes for
        performance
        reasons and triggers assertions in case of incompatible shapes. Please
            pad
        the input shapes accordingly.

    Returns:
        A unary function implementing the algorithm that takes the input array
            and
        returns a tuple of TopK values and indices with the shape ([batch_size
            ,
        num_elements], [batch_size, num_elements]) where num_elements is `
            local_K *
        num_buckets`.
    """
```

```
29    # Pallas imposes constraints on block specifications. Refer to the docs.
30    PALLAS_TPU_BLOCKSPEC_MAJOR_MULTIPLE = 8
31    PALLAS_TPU_BLOCKSPEC_MINOR_MULTIPLE = 128
32
33    input_shape = inputs.shape
34    batch_size, reduction_dims = input_shape
35    num_elements = num_buckets * local_K
36    output_shape = (batch_size, num_elements)
37
38    batch_tile_size = tunable_params.get('batch_tile_size', None)
39    if batch_tile_size is None:
40      factors = get_all_factors(batch_size)
41      legal_factors = {
42          f for f in factors
43            if f % PALLAS_TPU_BLOCKSPEC_MAJOR_MULTIPLE == 0
44            or f == batch_size
45      }
46      # Higher tile size provide more opportunities for instruction-level
47      # parallelism.
48      batch_tile_size = max({ f for f in legal_factors if f <= 2048 })
49    assert(batch_size % batch_tile_size == 0)
50
51    reduction_tile_size = tunable_params.get('reduction_tile_size', None)
52    if reduction_tile_size is None:
53      # In each subprogram, we would want to process sufficient number of
54            inputs
54      # to cover all the buckets. We would also want to have several passes
            over
55      # the buckets in each subprogram so that the compiler can schedule the
            loop
56      # iterations in a way that state loads/stores to the same buckets run
57      # consecutively, and the state information is cached in registers or
            in the
58      # nearest cache.
59      factors = set(get_all_factors(reduction_dims))
60      legal_factors = {
61          f for f in factors
62            if f % num_buckets == 0 and
63              f % PALLAS_TPU_BLOCKSPEC_MINOR_MULTIPLE == 0
64      }
65      # We want to pick sufficiently large blocks so that the overheads are
66      # amortized. However, we don't want the blocks to be too large to a
            point
67      # that we have too few pipelined iterations and the head and tail
            latencies
68      # make up a substantial portion of the runtime. These numbers vary
            from chip
69      # to chip and need to be tuned.
70      reduction_tile_size = max({
71          d for d in legal_factors if d <= max(32_768, num_buckets)
72      })
73    assert(reduction_dims % reduction_tile_size == 0)
74
```

```
75    # For simplicity , we restrict the tile sizes to be such that all the
          buckets
76    # are processed equal number of times in each subprogram.
77    assert(reduction_tile_size % num_buckets == 0)
78
79    input_transform_indices_fn = lambda i, j: (i, j)
80    input_tile_shape = (batch_tile_size , reduction_tile_size)
81    iteration_bounds = [
82        asz // tsz for asz, tsz in zip(input_shape , input_tile_shape)
83    ]
84    assert(len(input_tile_shape) == len(input_shape))
85
86    # Pallas currently does not allow non-consecutive grid points to write
          to the
87    # same slices of output. We therefore do not block the output along the
88    # reduction axis.
89    output_transform_indices_fn = lambda i, j: (i, 0)
90    output_tile_shape = (batch_tile_size , num_elements)
91    assert(len(output_tile_shape) == len(output_shape))
92
93    # At the time of writing , the Mosaic compiler does not have a rule for
94    # lowering comparisons between types that are not 32-bits wide. Hence ,
          we
95    # explicitly promote inputs to their wider 32-bit type.
96    assert(inputs.dtype.itemsize <= 4)
97    if jnp.issubdtype(inputs.dtype , jnp.floating):
98      compute_type = jnp.float32
99    elif jnp.issubdtype(inputs.dtype , jnp.signedinteger):
100     compute_type = jnp.int32
101   elif jnp.issubdtype(inputs.dtype , jnp.unsignedinteger):
102     compute_type = jnp.uint32
103   else:
104     assert "Unknown data type"
105
106   def _kernel(inputs_ref , values_ref , indices_ref):
107     assert(values_ref.shape == indices_ref.shape)
108
109     # b = batch axis , r = reduction axis
110     tile_b, tile_r = pl.program_id(0), pl.program_id(1)
111
112     # On TPUs , we are guaranteed a sequential grid execution and we use
            the
113     # first run for each batch to initialize the outputs.
114     @pl.when(tile_r == 0)
115     def initialize_outputs():
116       # We don't have to initialize the indices as non-strict comparators
              for
117       # selection guarantee that the indices will be updated.
118       values_ref[...] = jnp.full_like(values_ref , -jnp.inf)
119
120     # The loop count may be large but we explicitly want to unroll to
            eliminate
121     # a lot of state loads/stores. Rewriting the loop as two nested loops
            where
```

```
122    # the unrolled inner loop explicitly reuses the state load/stores and
           the
123    # non-unrolled outer loop runs over different sets of buckets may lead
           to
124    # faster compilation.
125    num_iterations_over_outputs = reduction_tile_size // num_buckets
126    for iter_idx in range(num_iterations_over_outputs):
127      # Note that inputs are already tiled by pallas and we use local
             offsets.
128      inputs = inputs_ref[
129          :, pl.ds(start=iter_idx * num_buckets, size=num_buckets)
130      ]
131      inputs = inputs.astype(compute_type)
132
133      iota = jax.lax.broadcasted_iota(indices_ref.dtype, inputs.shape, 1)
134      iota += tile_r * reduction_tile_size + iter_idx * num_buckets
135      assert(inputs.shape == iota.shape)
136
137      # Load state information for the current chunk.
138      values_by_k, indices_by_k = [], []
139      for k in range(local_K):
140        values = values_ref[
141            :,
142            pl.ds(
143                start=k * num_buckets,
144                size=num_buckets
145            )
146        ].astype(compute_type)
147        indices = indices_ref[
148            :,
149            pl.ds(
150                start=k * num_buckets,
151                size=num_buckets
152            )
153        ]
154        assert(values.shape == indices.shape)
155        values_by_k.append(values)
156        indices_by_k.append(indices)
157
158      # Compute the new state information for the current chunk.
159      pred = inputs >= values_by_k[-1]
160      values_by_k[-1] = jax.lax.select(pred, inputs, values_by_k[-1])
161      indices_by_k[-1] = jax.lax.select(pred, iota, indices_by_k[-1])
162      for k in reversed(range(1, local_K)):
163        # Note that the commented line and uncommented line are
             algorithmically
164        # equivalent, but the uncommented version has one less loop-
             carried
165        # dependency.
166        # pred = values_by_k[k] > values_by_k[k - 1]
167        pred = inputs > values_by_k[k - 1]
168
169        values_to_shift = values_by_k[k]
170        values_by_k[k] = \
```

```
171            jax.lax.select(pred, values_by_k[k-1], values_to_shift)
172          values_by_k[k-1] = \
173            jax.lax.select(pred, values_to_shift, values_by_k[k-1])
174
175          indices_to_shift = indices_by_k[k]
176          indices_by_k[k] = \
177            jax.lax.select(pred, indices_by_k[k-1], indices_to_shift)
178          indices_by_k[k-1] = \
179            jax.lax.select(pred, indices_to_shift, indices_by_k[k-1])
180
181      # Write the new state information for the current chunk.
182      for k in range(local_K):
183        values_ref[
184            :,
185            pl.ds(
186                start=k * num_buckets,
187                size=num_buckets
188            )
189        ] = values_by_k[k].astype(values_ref.dtype)
190        indices_ref[
191            :,
192            pl.ds(
193                start=k * num_buckets,
194                size=num_buckets
195            )
196        ] = indices_by_k[k]
197
198    def wrapper(inputs):
199      pr_values, pr_indices = pl.pallas_call(
200          _kernel,
201          in_specs=[
202              pl.BlockSpec(input_tile_shape, input_transform_indices_fn),
203          ],
204          out_shape=[
205              jax.ShapeDtypeStruct(output_shape, inputs.dtype),
206              jax.ShapeDtypeStruct(output_shape, jnp.int32),
207          ],
208          out_specs=[
209              pl.BlockSpec(output_tile_shape, output_transform_indices_fn),
210              pl.BlockSpec(output_tile_shape, output_transform_indices_fn)
211          ],
212          grid=iteration_bounds,
213          compiler_params=pltpu.TPUCompilerParams(
214            dimension_semantics=("parallel", "arbitrary")
215          ),
216          **kwargs
217      )(inputs)
218      return pr_values, pr_indices
219    return wrapper

 1 def make_generalized_approx_topk(operand, num_buckets, local_K, global_K,
      **kwargs):
 2    partial_reduce_fn = \
 3      generalized_partial_reduce(operand, local_K, num_buckets, **kwargs)
```

```
4
5     def wrapper(operand):
6       bucket_values, bucket_indices = partial_reduce_fn(operand)
7       values, indices = \
8         jax.lax.sort_key_val(bucket_values, bucket_indices, is_stable=
             False)
9       values = jnp.flip(values[..., -global_K:], axis=-1)
10      indices = jnp.flip(indices[..., -global_K:], axis=-1)
11      return values, indices
12    return wrapper
```

### A.9   Matmul-fused implementation of our algorithm.

```
1   def matmul_fused_generalized_partial_reduce(
2       lhs, rhs,
3       local_K, num_buckets,
4       tunable_params={}, *, **kwargs
5   ):
6     """Fused ApproxTopK with generalized partial reduce for minor-axis
         reductions.
7
8     Note: Input elements separated by a fixed stride form a bucket.
9
10    Args:
11      lhs: jax.ShapeDtypeStruct-compatible object of LHS array with shape
12        [batch_size, contracting_dims].
13      rhs: jax.ShapeDtypeStruct-compatible object of RHS array with shape
14        [contracting_dims, reduction_dims].
15      local_K: number of top elements to keep track of per bucket.
16      num_buckets: number of buckets.
17      tunable_params: These are hardware-specific (auto)tunable parameters.
           See
18        the uses in the function body for more information.
19      **kwargs: forwarded as is to `pallas_call`.
20
21    Note: The default tunable params selection does not check if the choices
           are
22      meaningful (e.g., VMEM OOMs). Please tune the choices if they don't
           work.
23
24    Note: The procedure does not allow arbitrary input shapes for
         performance
25      reasons and triggers assertions in case of incompatible shapes. Please
           pad
26      the input shapes accordingly.
27
28    Returns:
29      A binary function implementing the algorithm that takes the arguments
           to
30      matmul and returns a tuple of TopK values and indices of lhs @ rhs
           with
31      shapes of ([batch_size, num_elements], [batch_size, num_elements]).
32      The `num_elements` is calculated as `num_buckets * local_K`.
33    """
```

```
34
35    # Pallas imposes constraints on block specifications. Refer to the docs.
36    PALLAS_TPU_BLOCKSPEC_MAJOR_MULTIPLE = 8
37    PALLAS_TPU_BLOCKSPEC_MINOR_MULTIPLE = 128
38
39    # This implementation maps buckets to the minormost axis.
40    assert(num_buckets % PALLAS_TPU_BLOCKSPEC_MINOR_MULTIPLE == 0)
41
42    batch_size, contracting_dims = lhs.shape
43    contracting_dims_rhs, reduction_dims = rhs.shape
44    assert(contracting_dims == contracting_dims_rhs)
45    assert(reduction_dims % num_buckets == 0)
46    assert(num_buckets < reduction_dims)
47    assert(lhs.dtype == rhs.dtype)
48
49    num_elements = num_buckets * local_K
50    output_shape = (batch_size, num_elements)
51
52    # We will block the matrices for software pipelining as follows:
53    # lhs: [batch_tile_size, contracting_tile_size]
54    # rhs: [contracting_tile_size, reduction_tile_size]
55    # result-scratch: [batch_tile_size, reduction_tile_size]
56    #
57    # Note that partial reduce computation can start only after the loop
        over the
58    # contracting axis ends, as it requires fully accumulated sums to begin.
59    # The VPU may idle waiting for the result tile in all but the last
        iteration
60    # of the loop over the contracting axis.
61    #
62    # We can alleviate the problem by pipelining the computation into matmul
         and
63    # TopK stages. We let the VPU processes the previous result tile while
        the new
64    # result tile is being accumulated. We do not implement the idea here
        and
65    # choose to use large tiling along contracting axis to minimize wasted
        cycles.
66
67    # The tiles will be further subtiled automatically by the compiler to
        meet the
68    # shape of the hardware matmul units. Since the subtiling loops will be
        fully
69    # unrolled, the compiler would ideally generate code to run TopK on the
70    # previous subtile while a new subtile is being accumulated.
71
72    batch_tile_size = tunable_params.get('batch_tile_size', None)
73    if batch_tile_size is None:
74      # We want this to be as large as possible. This parameter controls the
75      # arithmetic intensity of the blocked matmul operation. Therefore, we
           must
76      # have a batch tile size that is high enough to ensure that each
           blocked
77      # matmul operation is MXU-bound.
```

```
78      factors = get_all_factors(batch_size)
79      legal_factors = {
80          f for f in factors
81            if f % PALLAS_TPU_BLOCKSPEC_MAJOR_MULTIPLE == 0
82            or f == batch_size
83      }
84      # We heuristically pick the largest legal tile size up to 2048. A
            larger
85      # tile size may be more performant but carries the risk of exhausting
            VMEM.
86      batch_tile_size = max({ f for f in legal_factors if f <= 2048 })
87    assert(batch_size % batch_tile_size == 0)
88
89    contracting_tile_size = tunable_params.get('contracting_tile_size', None
          )
90    if contracting_tile_size is None:
91      # This tile size does not affect the arithmetic intensity of the
            matrix
92      # multiplication. However, as mentioned earlier, we cannot start the
            TopK
93      # computation without the fully accumulated result tile. To minimize
            VPU
94      # idle time, we would like to have as large a tile size as possible
            for the
95      # contracting axis so that there are as few iterations as possible
            where the
96      # final result tile is only partially accumulated.
97      factors = get_all_factors(contracting_dims)
98
99      # This axis would be the minor axis for LHS and the major axis for RHS
            . It
100     # must meet the multiple requirements for the LHS and RHS respectively
             or
101     # must be equal to the axis size.
102     legal_factors = {
103         f for f in factors
104           if (f % PALLAS_TPU_BLOCKSPEC_MAJOR_MULTIPLE == 0 and
105               f % PALLAS_TPU_BLOCKSPEC_MINOR_MULTIPLE == 0)
106           or f == contracting_dims
107     }
108
109     # We heuristically pick the largest size up to 2048. A larger tile
            size
110     # would minimize VPU idling for this implementation but increases the
            risk
111     # of VMEM OOMs.
112     contracting_tile_size = max({ f for f in legal_factors if f <= 2048 })
113   assert(contracting_dims % contracting_tile_size == 0)
114
115   reduction_tile_size = tunable_params.get('reduction_tile_size', None)
116   if reduction_tile_size is None:
117     # There are two possibilities for picking reduction tile size:
118     # 1. tile size > number of buckets
119     # 2. tile size <= number of buckets
```

```
120        #
121        # Our implementation handles both cases. However, the first
              possibility is
122        # preferred for performance reasons.
123
124        # We need to ensure that we have sufficient VMEM to accommodate the
              large
125        # tiles for matrix multiplication so that it remains MXU-bound. Let's
              cap
126        # the tile size to 4096 to reduce the risk of exhausting VMEM.
127        if num_buckets > 4096:
128          assert(reduction_dims % num_buckets == 0)
129          factors = set(get_all_factors(num_buckets))
130          legal_factors = {
131              f for f in factors
132                if f % PALLAS_TPU_BLOCKSPEC_MINOR_MULTIPLE == 0
133          }
134          reduction_tile_size = max({ d for d in legal_factors if d <= 4096 })
135        else:
136          # We want to pick sufficiently large tiles so that the load/store
              overhead
137          # of the TopK lists is amortized. However, we don't want the blocks
              to be
138          # too large to a point that we have too few pipelining iterations
              and the
139          # head and tail latencies make up a substantial portion of the
              runtime.
140          assert(reduction_dims % num_buckets == 0)
141          factors = set(get_all_factors(reduction_dims))
142          legal_factors = {
143              f for f in factors
144              if f % num_buckets == 0
145              and f % PALLAS_TPU_BLOCKSPEC_MINOR_MULTIPLE == 0
146          }
147          reduction_tile_size = max({ f for f in legal_factors if f <= 4096 })
148
149      assert(reduction_dims % reduction_tile_size == 0)
150      if reduction_tile_size > num_buckets:
151        # For simplifying the implementation, we restrict the tile size to be
152        # multiples of number of buckets.
153        assert(reduction_tile_size % num_buckets == 0)
154      else:
155        # For simplifying the implementation, we restrict the tile size to be
156        # factors of number of buckets.
157        assert(num_buckets % reduction_tile_size == 0)
158
159      lhs_transform_indices_fn = lambda i, j, k: (i, k)
160      lhs_tile_shape = (batch_tile_size, contracting_tile_size)
161      assert(len(lhs_tile_shape) == len(lhs.shape))
162
163      rhs_transform_indices_fn = lambda i, j, k: (k, j)
164      rhs_tile_shape = (contracting_tile_size, reduction_tile_size)
165      assert(len(rhs_tile_shape) == len(rhs.shape))
166
```

```
167    result_tile_shape = (batch_tile_size, reduction_tile_size)
168
169    iteration_bounds = [asz // tsz for asz, tsz in zip(
170        [batch_size, reduction_dims, contracting_dims],
171        [batch_tile_size, reduction_tile_size, contracting_tile_size]
172    )]
173    contraction_steps = iteration_bounds[2]
174
175    # Pallas currently does not allow non-consecutive grid points to write
           to the
176    # same slices of output. We therefore do not block the output along the
177    # reduction axis.
178    output_transform_indices_fn = lambda i, j, k: (i, 0)
179    output_tile_shape = (batch_tile_size, num_elements)
180    assert(len(output_tile_shape) == len(output_shape))
181
182    # At the time of writing, the Mosaic compiler does not have a rule for
183    # lowering comparisons between types that are not 32-bits wide. Hence,
           we
184    # explicitly promote inputs to their wider 32-bit type.
185    assert(lhs.dtype.itemsize <= 4)
186    if jnp.issubdtype(lhs.dtype, jnp.floating):
187      compute_type = jnp.float32
188    elif jnp.issubdtype(lhs.dtype, jnp.signedinteger):
189      compute_type = jnp.int32
190    elif jnp.issubdtype(lhs.dtype, jnp.unsignedinteger):
191      compute_type = jnp.uint32
192    else:
193      assert "Unknown data type"
194
195    def _kernel(lhs_ref, rhs_ref, values_ref, indices_ref, acc_ref):
196      # b = batch axis, r = reduction axis, c = contracting axis
197      tile_b, tile_r, tile_c = \
198        pl.program_id(0), pl.program_id(1), pl.program_id(2)
199
200      if contraction_steps > 1:
201        @pl.when(tile_c == 0)
202        def reset_accumulators():
203          acc_ref[...] = jnp.zeros_like(acc_ref)
204
205        # For each output tile, we reset the accumulators to zero.
206        @pl.when(tile_c < contraction_steps - 1)
207        def matmul_only_step():
208          acc_ref[...] += jnp.matmul(
209              lhs_ref[...], rhs_ref[...], preferred_element_type=jnp.float32
210          )
211
212      # When we've accumulated all the partial products, we update the top-K
           ,
213      # lists with the new elements.
214      @pl.when(tile_c == contraction_steps - 1)
215      def update_topk_state():
216        assert(values_ref.shape == indices_ref.shape)
217
```

```
218        @pl.when(tile_r == 0)
219        def initialize_outputs():
220          # We don't have to initialize the indices as non-strict
                 comparators for
221          # selection guarantee that the indices will be updated.
222          values_ref[...] = jnp.full_like(values_ref, -jnp.inf)
223
224        if contraction_steps == 1:
225          acc_ref[...] = jnp.zeros_like(acc_ref)
226
227        acc_ref[...] += jnp.matmul(
228            lhs_ref[...], rhs_ref[...], preferred_element_type=jnp.float32
229        )
230
231        def update_state(inputs, iota, state_offset, state_size):
232          """Update the top-K' lists with the new inputs."""
233
234          assert(inputs.shape == iota.shape)
235          assert(inputs.shape[-1] == state_size)
236          assert(state_size <= num_buckets)
237
238          # Load state information for the current chunk.
239          values_by_k, indices_by_k = [], []
240          for k in range(local_K):
241            values = values_ref[
242                :,
243                pl.ds(
244                    start=pl.multiple_of(k * num_buckets + state_offset,
                          128),
245                    size=state_size
246                )
247            ].astype(compute_type)
248            indices = indices_ref[
249                :,
250                pl.ds(
251                    start=k * num_buckets + state_offset,
252                    size=state_size
253                )
254            ]
255            assert(values.shape == indices.shape)
256            values_by_k.append(values)
257            indices_by_k.append(indices)
258
259          # Compute the new state information for the current chunk.
260          pred = inputs >= values_by_k[-1]
261          values_by_k[-1] = jax.lax.select(pred, inputs, values_by_k[-1])
262          indices_by_k[-1] = jax.lax.select(pred, iota, indices_by_k[-1])
263          for k in reversed(range(1, local_K)):
264            # The commented line and uncommented line are algorithmically
265            # equivalent, but the uncommented version has one less loop-
                   carried
266            # dependency.
267            # pred = values_by_k[k] > values_by_k[k - 1]
268            pred = inputs > values_by_k[k - 1]
```

```
269
270            values_to_shift = values_by_k[k]
271            values_by_k[k] = \
272              jax.lax.select(pred, values_by_k[k-1], values_to_shift)
273            values_by_k[k-1] = \
274              jax.lax.select(pred, values_to_shift, values_by_k[k-1])
275
276            indices_to_shift = indices_by_k[k]
277            indices_by_k[k] = \
278              jax.lax.select(pred, indices_by_k[k-1], indices_to_shift)
279            indices_by_k[k-1] = \
280              jax.lax.select(pred, indices_to_shift, indices_by_k[k-1])
281
282        # Write the new state information for the current chunk.
283        for k in range(local_K):
284          values_ref[
285              :,
286              pl.ds(
287                  start=k * num_buckets + state_offset,
288                  size=state_size
289              )
290          ] = values_by_k[k].astype(values_ref.dtype)
291          indices_ref[
292              :,
293              pl.ds(
294                  start=k * num_buckets + state_offset,
295                  size=state_size
296              )
297          ] = indices_by_k[k]
298
299      if reduction_tile_size > num_buckets:
300        assert(reduction_tile_size % num_buckets == 0)
301        num_iterations_over_outputs = reduction_tile_size // num_buckets
302
303        # The loop count may be large but we explicitly want to unroll to
304        # eliminate a lot of state loads/stores.
305        for iter_idx in range(num_iterations_over_outputs):
306          # The inputs are already tiled by pallas and we use local
                 offsets.
307          inputs = acc_ref[
308              :, pl.ds(start=iter_idx * num_buckets, size=num_buckets)
309          ].astype(compute_type)
310
311          iota = jax.lax.broadcasted_iota(indices_ref.dtype, inputs.shape,
                 1)
312          iota += tile_r * reduction_tile_size + iter_idx * num_buckets
313          assert(inputs.shape == iota.shape)
314
315          update_state(inputs, iota, 0, num_buckets)
316      else:
317        assert(num_buckets % reduction_tile_size == 0)
318        inputs = acc_ref[...].astype(compute_type)
319
```

```
320        iota = jax.lax.broadcasted_iota(indices_ref.dtype, inputs.shape,
             1)
321        iota += tile_r * reduction_tile_size
322        assert(inputs.shape == iota.shape)

324        num_tiles_over_outputs = num_buckets // reduction_tile_size
325        state_offset = (tile_r % num_tiles_over_outputs) *
             reduction_tile_size

327        update_state(inputs, iota, state_offset, reduction_tile_size)

329   def wrapper(lhs, rhs):
330     pr_values, pr_indices = pl.pallas_call(
331       _kernel,
332       grid_spec=pltpu.PrefetchScalarGridSpec(
333         num_scalar_prefetch=0,
334         in_specs=[
335           pl.BlockSpec(lhs_tile_shape, lhs_transform_indices_fn),
336           pl.BlockSpec(rhs_tile_shape, rhs_transform_indices_fn),
337         ],
338         out_specs=[
339           pl.BlockSpec(output_tile_shape, output_transform_indices_fn),
340           pl.BlockSpec(output_tile_shape, output_transform_indices_fn)
341         ],
342         scratch_shapes=[pltpu.VMEM(result_tile_shape, jnp.float32)],
343         grid=iteration_bounds,
344       ),
345       out_shape=[
346         jax.ShapeDtypeStruct(output_shape, compute_type),
347         jax.ShapeDtypeStruct(output_shape, jnp.int32),
348       ],
349       compiler_params=pltpu.TPUCompilerParams(
350         dimension_semantics=("parallel", "arbitrary", "arbitrary")
351       ),
352       **kwargs
353     )(lhs, rhs)
354     return pr_values, pr_indices
355   return wrapper

  1 def make_matmul_fused_generalized_approx_topk(
  2     lhs, rhs, num_buckets, local_K, global_K, **kwargs
  3 ):
  4     partial_reduce_fn = \
  5       matmul_fused_generalized_partial_reduce(lhs, rhs, local_K,
           num_buckets, **kwargs)

  7     def wrapper(lhs, rhs):
  8       bucket_values, bucket_indices = partial_reduce_fn(lhs, rhs)
  9       values, indices = \
 10         jax.lax.sort_key_val(bucket_values, bucket_indices, is_stable=
             False)
 11       values = jnp.flip(values[..., -global_K:], axis=-1)
 12       indices = jnp.flip(indices[..., -global_K:], axis=-1)
 13       return values, indices
```

```
14      return wrapper
15
16  def matmul_fused_generalized_approx_topk(lhs, rhs, *args, **kwargs):
17      return make_matmul_fused_generalized_approx_topk(
18          lhs, rhs, *args, **kwargs)(lhs, rhs)
```

### A.10 Algorithm Parameter Selection

#### A.10.1 Monte Carlo Estimation of Expected Recall

```
1  def expected_recall_mc(N, B, K_global, K_local, num_trials):
2      assert(N % B == 0)
3      bucket_size = N // B
4      X_samples = np.random.hypergeometric(
5          K_global,
6          N - K_global,
7          bucket_size,
8          size=num_trials
9      )
10     num_collisions = B * np.maximum(X_samples - K_local, 0)
11     recall = 1 - num_collisions / K_global
12     expected_recall = np.mean(recall)
13     std_error = np.std(recall, ddof=1) / np.sqrt(num_trials)
14     return expected_recall, std_error
```

#### A.10.2 Parameter Sweep

```
1  def select_parameters(
2      input_size, K,
3      recall_target,
4      allowed_local_K=[1, 2, 3, 4]
5  ):
6      """Finds a good set of algorithm parameters for the given configuration.
7
8      Args:
9        input_size: size of the array
10       K: number of top entries
11       recall_target: minimum "expected" recall required
12       allowed_local_K: list of local K to consider in the search space
13     """
14
15     divisors = get_all_factors(input_size)
16     allowed_num_buckets = [ d for d in divisors if d % 128 == 0 ]
17
18     # For a fixed K, the expected recall decreases as the number of buckets
19     # decreases. Therefore, by sweeping through `num_buckets` in descending
20     # order, we can terminate the search early when we miss the recall
21     #     target.
22     allowed_num_buckets = sorted(allowed_num_buckets, reverse=True)
23
24     # The best configuration selection logic only checks for the total
25     #     number of
26     # elements using a strict comparison. We want to try local K in
27     #     ascending
```

```
25    # order so that in the case of a tie , we pick the configuration with a
          smaller
26    # local K.
27    allowed_local_K = sorted ( allowed_local_K )
28
29    best_config = None
30    best_num_elements = np.inf
31    for local_K in allowed_local_K :
32      for num_buckets in allowed_num_buckets :
33        if num_buckets * local_K < K :
34          break
35
36        if recall_target >= 0.995:
37          warnings.warn (
38              f"recall_target of {recall_target} too high"
39              " for reliable selection of algorithm.",
40              RuntimeWarning
41          )
42
43        num_trials = 4096
44        recall , recall_err = \
45          expected_recall_mc ( input_size , num_buckets , K , local_K , num_trials
              )
46        while recall_err * 3 > 0.005:
47          num_trials *= 2
48          recall , recall_err = \
49            expected_recall_mc ( input_size , num_buckets , K , local_K ,
                num_trials )
50
51        if recall < recall_target :
52          break
53
54        num_elements = num_buckets * local_K
55        if num_elements < best_num_elements :
56          best_config = ( local_K , num_buckets )
57          best_num_elements = num_elements
58    assert ( best_config is not None )
59    return best_config
```

### A.10.3  Computational cost of parameter selection routine.

The parameter sweep evaluates configurations in descending order of bucket count for each value of $K'$, terminating early once the recall target is no longer met. This works since recall decreases monotonically with fewer buckets. For each configuration evaluated, we use an adaptive sampling procedure that draws hypergeometric samples until the recall estimate is within $\pm 0.005$ at 3-sigma confidence. To give a sense of the cost, we evaluated eight representative configurations spanning array sizes from 16k to 917k and K from 128 to 3360 at 95% recall target; the parameter sweep evaluated between a few dozen and up to 115 configurations per (N, K, recall_target) combination, drawing between 930k and 5M hypergeometric samples in total, and completed in under a second on an AMD Ryzen 7 3700X desktop CPU. This cost is negligible relative to the typical compilation times for LLMs, which range from a few minutes to several hours. In practice, the selected parameters can be cached and reused across all calls with the same configuration, which is common in LLMs where repeated identical blocks of layers dominate the architecture. The implementation has not been optimized; straightforward improvements such as binary search over bucket counts or sharing random samples across configurations could further reduce the cost.

### A.10.4  Discussion and Generalization

The parameter sweep is currently specific to TPUv5e. The model in Section 2.3 and Table 1 determines the maximum $K'$ for which the first stage remains memory-bound and therefore costs approximately the same regardless of $K'$ (confirmed empirically in Table 2, where Stage 1 latency is nearly flat from $K' = 1$ to $K' = 6$). Within this regime, minimizing the number of output elements ($B \times K'$) is equivalent to minimizing total latency, so the sweep correctly optimizes the right objective without explicitly consulting Table 1 at search time. On TPUv5e, the ridge point analysis gives $K' = 6$ as the crossover between memory-bound and compute-bound first-stage behavior; we conservatively restrict to $K' \leq 4$ to leave headroom for other operations like casting. The constraint that the number of buckets be a multiple of 128 and a divisor of the input size arises from TPUv5e-specific alignment requirements in our implementation.

The underlying methodology nonetheless generalizes. For any accelerator for which the quantities in Table 1 can be measured or obtained from the datasheets, one can: (i) compute the ridge point to determine the $K'$ ceiling below which the first stage remains memory-bound; (ii) identify the implementation constraints; and (iii) run the same sweep to find the configuration minimizing second-stage input size. We can in principle extend this existing sweep routine to other TPUs directly using corresponding numbers from Table 1.

The benefits of hardware-aware parameter selection are most significant in fused settings. When operations such as ReLU, sigmoid, or elementwise products (e.g., in GLU-based MLP blocks) are fused alongside the Top-$K$ first stage, they compete for VPU instructions and reduce the effective vector compute budget available for Top-$K$, thereby lowering the effective $K'$ ceiling. Conversely, when fusing with matrix multiplications with large contracting dimensions, substantially more VPU compute becomes available than the baseline estimate from Section 2.3 suggests (as discussed in Section 5), potentially enabling $K'$ well beyond 6. A complete fusion-aware parameter selection procedure would account for these competing demands when determining the $K'$ ceiling, and could in principle be integrated into a compiler cost model to automate fusion decisions. This framework applies broadly, not just to matmul fusions, but to any kernel with which Top-$K$ might be fused, including memory-bound kernels.

## A.11 Expected recall rapidly improves with increasing $K'$.

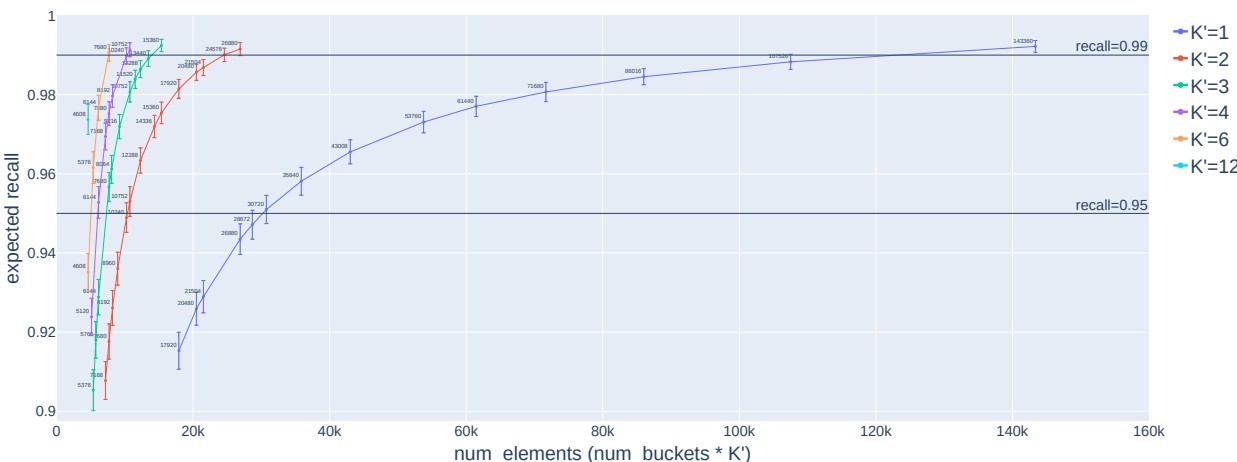

Figure 10: **Recall vs number of elements for finding top-3360 ($\approx 0.8\%$) elements from an array of size 430,080.** The data was obtained from simulated runs of the algorithm on randomly generated integers. The markers represent the sample mean and the error bars represent the sample standard deviation from 1024 trials. Each curve corresponding to a $K'$ depicts the Pareto frontier for that $K'$. The ideal point is $(K, 1.0)$. Beyond a certain $K'$, the first stage will become sufficiently expensive that the additional cost of the first stage exceeds the gains in the second stage. However, for small values of $K'$, where the additional cost of the first stage is negligible, we note that the Pareto frontier improves as $K'$ increases.

## A.12 Arithmetic intensity of fused matmul operation.

$$
\begin{aligned}
\text{arithmetic intensity} &= \frac{2BDN}{E\left(BD + DN + BN\right)} \\
&\approx \frac{2BDN}{E\left(DN + BN\right)} \\
&= \frac{2BD}{E\left(B + D\right)} \\
&\leq \frac{2}{E}\min(B, D).
\end{aligned}
$$

## A.13 Training for sparse activations in MLP blocks.

We benchmark a non-gated MLP variant of Gemma 2 9B using SquaredReLU activations to induce sparsity, following prior work on activation sparsity (Samaga B L et al., 2024; Mirzadeh et al., 2023; Zhang et al., 2024). To maintain rough parameter parity with the gated baseline, we set the intermediate dimension in the MLP blocks to 24,576, while keeping all other architectural hyperparameters unchanged. We use a sequence length of 1024 and a per-rank batch size of 8. To enforce sparsity, we apply a Top-K algorithm that selects approximately the top 2% of FFN activations (K = 512 out of 24,576, i.e., 2.08%), targeting 95% recall.

We profile isolated residual blocks (including both forward and backward passes). In the dense baseline, an MLP residual block requires 33ms, while an attention block requires 16ms. Using the Top-K method of Chern et al. (2022), the sparse MLP block takes 89ms, which is approximately 2.7× slower than dense MLP. This overhead causes the Top-K operation to dominate not only the MLP computation but also the overall transformer block (MLP + attention ≈ 50ms in the dense setting). In contrast, our method increases the sparse MLP latency to only 38 ms, representing a modest +5 ms overhead relative to the dense baseline.

