# OpenReview forum: "A Faster Generalized Two-Stage Approximate Top-K"
_TMLR — Accepted by TMLR_

### Review · Reviewer_F1SM · 2026-02-18

**Summary Of Contributions:**

This paper focuses on improving the efficiency of an approximaton of the top-$K$ operation over arrays on accelerators such as TPUs and GPUs. The proposed scheme builds upon the existing 2-stage approach of Chern et al. (2022), implemented as `jax.lax.approx_max_k`, where a first stage divides the input array into equal sized buckets (say $B$ buckets), computes their respective top-1 in parallel, and then collects the results, and a more computational demanding second stage computes the exact top-$K$ over the collected results. The number of buckets selected for the first stage drives the approximation.

Motivated by careful understanding of the level of parallelism available on accelerators (specifically a TPU) and probabilistic analysis of the approximation in the 2-stage approach, the authors consider a generalization of the approach of Chern et al. (2022) where the first stage splits the data in $B$ buckets and computes the exact top-$K'$ for each bucket in parallel. Then, in the second stage, the algorithm aggregates the results and computes the exact top-$K$  as before. The results show that there are scenarios where choosing a $K' > 1$ actually allows for smaller computational overhead in the second stage, while leveraging the parallelism available in the first stage more strategically (and fusing the first stage with matrix-multiplication which often precedes the top-$K$ operation).

The empirical results highlight the advantage of the generalized algorithm and the computational gains obtained by its implementation which automatically configures the choice of $K'$ and the number of buckets $B$ for the first stage based input (the input array size $N$, the choice of $K$ and the level of approximation allowed $r$).


Strengths:

- **[S1]** The top-$K$ operation appears in many scenarios now even in the specific context of LLMs, and thus, the problem of speeding up this operation on accelerators such as GPUs or TPUs can be quite impactful. The overall main contribution is a faster approximate top-$K$ algorithm and implementation with strong empirical performance.
- **[S2]** The paper presents a careful analysis of the failure probability which allows for a generalized algorithm that even improves the existing algorithm of Chern et al.(2022). Section 6.2 provides a nice analysis, and comparison to the analysis of Chern et al. (2022).
- **[S3]** This algorithm is tuned for a TPU implementation with careful alignment of the algorithm with the structural requirements of TPUs. The authors do a great job of explaining the relevant background regarding the computational model for accelerators, and tie in the ridge-point analysis (in Section 2.3) to the empirical results (such as the one in Section 7.3).



Weaknesses:

- **[W1]** One of the main weaknesses is the existence of (potentially concurrent) work of Key et al (2024), which makes the novelty of the proposed scheme and analysis quite limited. The authors do address this in Section 4. However, I think it might be beneficial to highlight (empirically) how the contributions of the current submission does more than Key et al. (2024) for example by evaluating their proposed (matmul+top-$K'$) fused version (on a GPU) to the open-source [pytorch-approx-topk implementation](https://github.com/graphcore-research/pytorch-approx-topk/). Furthermore, it might be useful to contextualize the analysis in Section 6.2 to the one in Key et al (2024) (Appendix D). To the best of my understanding, they are the same, and come to the same conclusion of tighter bounds even for $K' = 1$ than Chern et al. (2022).
- **[W2]** While the focus is on the approximate top-$K$ scheme of Chern et al. (2022), I believe some comparison to other fast top-$K$ schemes (discussed in Section 4) if possible would further strengthen the contributions of this paper.
- **[W3]** Given that this paper is not very tight on space (for example, page 11 just contains a single figure), it think that some space could be dedicated to a concluding section that highlights existing limitations and potential for future improvements.
- **[W4]** The implementation optimizes the configuration of the first stage $(K', B)$ based solely on the array length processed in the second stage $(B \times K')$ as shown in Appendix A.8. While this is a reasonable heuristic, this configuration choice is agnostic to the actual accelerator parameters such as the ones discussed in Section 2.3 and Table 1. It is not clear if they should play a role in the first-stage configuration choice. This could use an explicit discussion in the paper.
- **[W5]** While Key et al. (2024) study both $K \ll N$ and $K \propto N$ scenarios, it appears that this paper mostly focuses on the $K \ll N$ case without explicitly mentioning it. Figure 4 presents results for the $K \propto N$ scenario. However, from the results and the discussions, it is not clear, if for example, for a fixed $N$, whether increasing $K$ increases or decreases the computational gains, or for a fixed $K/N$ ratio, whether increasing $N$ increases or decreases the gains. This could use a deeper discussion as it would be insightful to a reader.

**Audience:**

Yes

**Audience Explanation:**

I think a large part of the TMLR audience would benefit from knowing the findings of this paper. This is because of the following reasons:
- The wide applicability of an accelerator-friendly top-$K$ operation in the context of LLMs and beyond makes the findings of this paper widely applicable.
- I think the authors provide a good discussion of the background on accelerators necessary for the community, which also highlights the contribution better.
- The availability of a top-$K$ operation that can be faster than the matmul itself (as shown in the results in Table 3) can allow the community to more widely and efficiently evaluate the benefits of top-$K$ in all aspects of the LLM pipeline, potentially allowing for inference on low-memory devices.
- An efficient implementation of top-$K$ also opens the door for a scalable evaluation of its advantages during training. For example, recent work have highlighted the gains of training convergence and generalization for top-$K$ attention transformers on compositional tasks, and the proposed approximate top-$K$ can allow one to scale these results to larger problems.

**Broader Impact Concerns:**

No broader impact statement and no concerns.

**Claims And Evidence:**

Yes

**Claims Explanation:**

Given the focus on the existing approximate top-$K$ algorithm of Chern et al. (2022), the paper presents sufficient evidence, both theoretical and empirical, to demonstrate that the proposed new algorithm provides significant computational savings over the existing scheme. Section 5 and 6 clearly support the claims at a theoretical level, while section 7 provides clear empirical evidence.

There is of course room for improvement as mentioned in weaknesses **[W1]** and **[W2]**, which highlight the lack of a wider evaluation beyond comparison to Chern et al. (2024). Furthermore, the paper lacks any evaluation pertaining to downstream runtimes and performance for tasks such as $K$-nearest-neighbor or $K$-MIPS recall or LLM prediction (where some part of the LLM pipeline makes use of the top-$K$ operation) as considered in Key et al. (2024) (Figure 2, right).

**Requested Changes:**

While **[W1]** is an important issue that should be addressed, I feel that it might require nontrivial effort potentially rendering it out-of-scope. Thus, I leave that to the discretion of the editor. However, I would really like to see some discussion regarding **[W3]**, **[W4]** and **[W5]**.

Beyond the weaknesses, here are some other potentually minor changes:

- Equation (1) is not properly introduced. For example, we do not know what $r$ is.
- In Figure 4, and the corresponding discussion on Page 10, the value of $K$, an integer, is provided as $K \in \lbrace 0.1, \ldots, 25 \rbrace$ percentages. I am guessing this is meant to be $K \in \lbrace 0.1 N .100, \ldots, 25 N / 100 \rbrace $, and it would be good to explicitly present it that way.
- For Alg 1, it is not clear if we need a `break` after line 12 if the `if values[k] > values[k-1] then` condition for lines 9-12 fails as we are assuming that the lists are always sorted pre-insertion as per the precondition, Furthermore, it seems that we would need the bubble sort loop in lines 8-13 to be executed only in case of an insertion when the `if input >= values[K'] then` holds; otherwise we could return at that point. Is this an oversight, or intentional? Is the algorithm set up this way to ensure that each `TopKPrimeUpdate` for each bucket takes the same amount of time.

---

> ### Author Response · Authors · 2026-03-28
> **Thank your for your comments - part 1.**
>
> > [W1] One of the main weaknesses is the existence of (potentially concurrent) work of Key et al (2024), which makes the novelty of the proposed scheme and analysis quite limited. The authors do address this in Section 4. However, I think it might be beneficial to highlight (empirically) how the contributions of the current submission does more than Key et al. (2024) for example by evaluating their proposed (matmul+top-) fused version (on a GPU) to the open-source pytorch-approx-topk implementation. Furthermore, it might be useful to contextualize the analysis in Section 6.2 to the one in Key et al (2024) (Appendix D). To the best of my understanding, they are the same, and come to the same conclusion of tighter bounds even for than Chern et al. (2022).
>
> We would like to point out that the initial version of this paper was submitted to MLSys conference in October 2024, while the Key et al. paper was uploaded to arxiv in December 2024. Nevertheless, we highlight the key differences of our paper compared to Key et al.
>
> **Theoretical results**: The recall expression derived in Appendix C of Key et al. is approximate and not exact. Specifically, since the algorithm distributes ‘n’ elements into ‘b’ buckets, the distribution of the number of elements in each bucket follows the hypergeometric distribution, not the binomial distribution. In fact, the derived recall expression is independent of ‘n’, which is not true without further assumptions on the values of k, b, k_b and n. As a consequence, their recall expression for the case k_b=1 does not depend on ‘n’ and is also approximate. The dependence on ‘n’ is indeed noticeable as can be seen from the difference in quadratic and quartic approximations in Figure 9 of our paper. In contrast, the expressions we derive are exact (Figure 6 and Figure 7) and the bounds we get out of them e.g., for K’=1 are essentially tight (for example, compare MC estimates and quartic expansion in Figure 9).
>
> **Systems contributions**:
> 1. **Performance modeling**: We generalize the instruction throughput-aware roofline model proposed in Chern et al. to work with arbitrarily many accelerator subsystems, and use this framework to design and implement our algorithm. We use the framework to identify the shortcomings of their algorithm in a principled manner, and in turn use the same framework to propose and motivate our generalization. We further instantiate this framework to select algorithm parameters in a principled manner and accurately predict runtime performance across different configurations, providing the foundation for building cost models that could support automatic compiler-driven fusion decisions. In contrast, Key et al. do not provide concrete theoretical performance modeling capable of predicting runtime, and instead rely on experiments for runtime estimation.
> 2. **Matmul fused kernels**: We present a matmul-fused version of the algorithm, important for performance on top-k similarity search workloads (see Appendix A12 and Section 4.2 in Chern et al.) as well as for extracting non-trivial gains in other settings. Key et al. does not discuss fusions.
>
>
> **Ease of use**: Our final interface directly accepts recall_target as an argument and is more intuitive: we use approx_top_k(array, K, recall_target) compared to Key et al.’s approx_top_k(array, K, K’). Our work enables automatic selection of K’ that meets the recall target (based on our theoretical results) and gives the best performance (runtime performance modeling). This eliminates the need for manual tuning of K’, as required in their work.

---

> ### Author Response · Authors · 2026-03-28
> **Thank you for your comments - part 2.**
>
> > [W2] While the focus is on the approximate top-K scheme of Chern et al. (2022), I believe some comparison to other fast top-K schemes (discussed in Section 4) if possible would further strengthen the contributions of this paper.
>
> Our algorithm is a generic wrapper that transforms any exact Top-K algorithm into an approximate one by adding a lightweight first stage. It therefore does not compete with exact algorithms; it subsumes and generalizes them. Since the first stage only reduces the input size for the second stage, our algorithm is always at least as fast as whatever exact algorithm is used in the second stage, at any given recall target. Furthermore, the motivation for approx. Top-K algorithms such as Chern et al. is precisely that exact Top-K is too slow for practical use on accelerators, and we’re directly improving upon this widely deployed baseline.
>
> $\space$
>
> > [W3] Given that this paper is not very tight on space (for example, page 11 just contains a single figure), it think that some space could be dedicated to a concluding section that highlights existing limitations and potential for future improvements.
>
> We've added a conclusion section in the revised version.
>
> *We present a generalization of the approximate Top-$K$ algorithm of \citet{chern2022tpuknnknearestneighbor} that selects the top-$K'$ elements per bucket in the first stage, instead of restricting to top-1. We theoretically analyze the expected recall of the generalized algorithm deriving an exact expression for it, and obtain a bound for $K’=1$ which is provably tighter by a factor of 2 compared to that in \citet{chern2022tpuknnknearestneighbor}. For $K'>1$, we demonstrate both theoretically and empirically that the algorithm substantially reduces the second-stage input size across a wide range of configurations, spanning $\frac{K}{N}$ ratios from 0.01\% to 25\%, with a median reduction of $7\times$ over the $K'=1$ baseline. We implement the algorithm for Cloud TPUv5e with a principled performance modeling framework that accurately predicts runtimes, enabling a user-friendly \texttt{approx\_top\_k(array, K, recall\_target)} interface that requires no manual tuning. We additionally present a matmul-fused implementation that further eliminates the first-stage cost. While our kernel implementations are specific to TPUs, the algorithm and its analysis are general and we believe that it has applicability to other hardware platforms such as GPUs.*
>
> *Limitations and Future Work. The empirical evaluation is focused on Cloud TPUv5e; extending empirical validation and the matmul-fused kernel to GPUs is an important open direction. The recall analysis assumes uniform random placement of top-$K$ elements; efficient parallel randomization schemes that bring real-world inputs in line with this model would fully close the gap between theory and practice. Integrating the algorithm into an optimizing compiler would enable automatic fusion decisions and fusion-aware parameter selection. Our generalization of the first stage represents the simplest step in a richer design space; more sophisticated first-stage selection schemes beyond the top-$K'$ per bucket approach could push the performance envelope further.*

---

> ### Author Response · Authors · 2026-03-28
> **Thank you for your comments - part 3.**
>
> > [W4] The implementation optimizes the configuration of the first stage based solely on the array length processed in the second stage as shown in Appendix A.8. While this is a reasonable heuristic, this configuration choice is agnostic to the actual accelerator parameters such as the ones discussed in Section 2.3 and Table 1. It is not clear if they should play a role in the first-stage configuration choice. This could use an explicit discussion in the paper.
>
> We would like to first note that the parameter sweep in A.8 is currently specific to TPUv5e. The model in Section 2.3 and Table 1 determines the maximum K′ for which the first stage remains memory-bound and therefore costs approximately the same regardless of K′ (confirmed empirically in Table 2, where Stage 1 latency is nearly flat from K′=1 to K′=6). Within this regime, minimizing the number of output elements from the first stage is equivalent to minimizing total latency, so the parameter sweep correctly optimizes the right objective without needing to re-consult Table 1 at search time. A more aggressive optimization that trades first-stage performance for second-stage performance would require building a full cost model for the first stage using Section 2.3 and Table 1. On TPUv5e specifically, the ridge point analysis gives K′=6 as the crossover; we conservatively allow up to K′=4 to leave headroom. The other constraints in the sweep, such as limiting the number of buckets to multiples of K′ and multiples of 128, also arise from TPUv5e-specific implementation constraints. That said, the underlying methodology, computing the ridge point from Table 1, setting the K′ ceiling, and minimizing second-stage input size subject to hardware alignment constraints, generalizes to any accelerator for which the relevant quantities can be measured or obtained from datasheets (as done in Table 1). We can in principle extend the sweep to other TPUs by computing the K′ upper bound from Table 1.
>
> The real benefits of our modeling, as noted in Section 4, arise most acutely in fused settings. When operations such as ReLU or sigmoid activations are fused after the matrix multiplication (e.g., MLP blocks of transformers), they compete for VPU instructions, reducing the vector compute budget available for the Top-K first stage and thereby lowering the effective K′ ceiling. The same applies to GLU-based blocks, where the elementwise product and activation both contend for VPU compute. Furthermore, as our analysis in Section 5 highlights, Chern et al. underestimates the available VPU compute based on the Section 2.3 model: when fusing with matrix multiplications with large contracting dimensions, substantially more VPU compute becomes available, potentially allowing K′ well beyond 6. Explicitly accounting for these competing demands when selecting K′ could enable tighter, fusion-aware parameter selection. Note also that this framework is not limited to matmul fusions; Top-K fused with memory-bound kernels would benefit from the same treatment. While we do not yet have a compiler integration to demonstrate automated fusion decisions, we believe the performance modeling framework presented here provides a solid foundation for such automation, and we view compiler-driven fusion-aware parameter selection as a promising direction for future work. We will add an explicit discussion of all these points to Appendix A.8.

---

> ### Author Response · Authors · 2026-03-28
> **Thank you for your comments - part 4.**
>
> > [W5] While Key et al. (2024) study both and scenarios, it appears that this paper mostly focuses on the case without explicitly mentioning it. Figure 4 presents results for the scenario. However, from the results and the discussions, it is not clear, if for example, for a fixed , whether increasing increases or decreases the computational gains, or for a fixed ratio, whether increasing increases or decreases the gains. This could use a deeper discussion as it would be insightful to a reader.
>
> We thank the reviewer for this thoughtful question and agree this deserves clarification. We establish the picture with two key facts:
> 1. **Stage 1 cost is negligible.** Stage 1 reads N elements from memory and performs (5K′ − 2) VPU ops per element (Section 6.3). As long as K′ is below the VPU-HBM ridge point (≈6 on TPUv5e) as defined in Section 2.3, Stage 1 is memory-bound and its cost is dominated by the unavoidable cost of reading N elements and writing B·K′ output elements, which is upper bounded by N. Stage 2, by contrast, is substantially more expensive than a single sequential read of its inputs — exact Top-K on 65K elements costs nearly a hundred times more than simply reading 65K elements from memory (derivable from Tables 1 and 2). Therefore, Stage 1 is almost always negligible compared to Stage 2 for the configurations we care about, and runtime is dominated entirely by Stage 2. The only configurations where Stage 2 can be faster than Stage 1 are when B·K′ is so small relative to N that the unavoidable cost of reading N elements in Stage 1 exceeds the cost of exact Top-K on a very small number of elements — this occurs roughly when B·K′ ≲ 512 for the example in Table 2.
> 2. **K′ should always be maximized.** Since runtime is dominated by Stage 2, the goal reduces to minimizing B·K′ for a fixed recall target. Our theory shows that recall improves at a faster rate by increasing K′ than by increasing B at fixed B·K′, so the optimal strategy is always to increase K′ as much as possible and set B to the minimum needed to meet the recall target. Since we would like to keep Stage 1 memory-bound, we constrain K′ to a small number determined by the hardware constants (Table 1).
>
> One subtlety: our parameter selection searches over multiple values of K′, which may seem inconsistent with always preferring higher K′. This arises because B must be a multiple of 128 and divide N for our implementation, and rounding up can occasionally make a smaller K′ with a better-aligned B result in fewer total B·K′ elements. There are means to reduce these effects such as searching over a space that includes padding for N to a number with many factors or powers of two.
>
> The gains in reality are essentially continuous with K and N as approximately shown in Figure 4, rather than falling into two discrete regimes. The discontinuities visible in Figure 4 are rounding artifacts. A neat analytical treatment of these rounding effects would be interesting, but appears nontrivial.
>
> The regime bifurcation in Key et al. is likely an artifact of their implementation constraints, similar in spirit to our 128-bucket-multiple constraint, rather than a fundamental property of the algorithm. Based on our analysis, for small K' ($\le 4$), we expect their first stage to also be memory-bound on A100s & H100s (see Table 1), and it is therefore unclear to us whether their claim of increasing B to expose more parallelism in $K \ll N$ regime provides any benefit. Our parameter selection routine does sometimes select a lower K′ just like theirs in $K \ll N$ regime, but this arises exclusively from rounding effects rather than a principled regime-based argument. In that sense, the regime bifurcation does not apply in our case as a fundamental distinction.
>
> $\space$
>
> > Beyond the weaknesses, here are some other potentually minor changes: ...
>
> We've addressed these in the revised version.
>
> $\space$
>
> > For Alg 1, it is not clear if we need a break after line 12 if the if values[k] > values[k-1] then condition for lines 9-12 fails as we are assuming that the lists are always sorted pre-insertion as per the precondition, Furthermore, it seems that we would need the bubble sort loop in lines 8-13 to be executed only in case of an insertion when the if input >= values[K'] then holds; otherwise we could return at that point. Is this an oversight, or intentional? Is the algorithm set up this way to ensure that each TopKPrimeUpdate for each bucket takes the same amount of time.
>
> This is intentional to allow vectorization over buckets. We've added the following clarification in Section 6.3.
>
> *Note that Algorithm 1 does not include an early return if the condition on Line 4 fails, nor does it exit the bubble sort loop early when the condition in Line 9 fails. This is required to vectorize the routine across buckets. An early return would theoretically reduce operations for individual buckets but would prevent vectorization across buckets.*

---

> > ### Comment · Reviewer_F1SM · 2026-04-09
> > **Acknowledging the author response**
> >
> > I thank the authors for the additional discussion and the corresponding updates to the submission:
> >
> > - Thank you for your discussion regarding **[W1]**. As I mentioned, I did not hold it against the submission. But I think the additional discussion positions the submission better with existing prior art.
> > - Thank for the new conclusion section, and the discussion of limitations and future work. I think it wraps up the paper nicely.
> > - Thank you for clarifying my misunderstanding regarding **[W4]** as I did not understand that the automatic parameter selection was hardware specific, where the hardware parameters are utilized to compute the ridge point.
> > - The explanation regarding **[W5]** is very helpful.

---

### Review · Reviewer_ZuW6 · 2026-03-01

**Summary Of Contributions:**

This paper generalizes the two-stage approximate Top-K algorithm of Chern et al. (2022) — currently deployed in JAX as jax.lax.approx_max_k — by allowing the first stage to select the top-$K'$ elements per bucket (rather than just top-1). The key contributions are:

*Tighter theoretical analysis*

The paper's main theoretical contribution is an exact expected recall expression for the generalized two-stage approximate Top-K algorithm, derived by modeling bucket occupancy with the hypergeometric distribution --- the correct model for partitioning without replacement, unlike the birthday-problem approximation used by Chern et al. (2022). From this expression, the authors prove that for $K'=1$, the number of buckets needed to guarantee a recall target $r$ is $B = K / (2(1-r+K/(2N)))$, which is provably 2x tighter than Chern et al.'s $B = K/(1-r)$. The tighter bound arises from properly accounting for the fact that one colliding element per bucket is always retained, which Chern et al. overlooked. For general $K'>1$, the exact expression enables a parameter sweep to find the $(K', B)$ pair minimizing the total first-stage output size $B \cdot K'$ subject to a recall constraint, revealing that modest values of $K' (2–4)$ can reduce the second-stage input by a median factor of 7x.

*Algorithmic generalization*

They show that for $K'>1$, substantially fewer total elements need to be passed to the expensive second-stage sort while maintaining the same expected recall — a median 7x reduction across configurations (Figure 4).

*TPU implementation*

They provide a Pallas-based implementation for TPUv5e (unfused and matmul-fused), achieving ~10x speedup over jax.lax.approx_max_k with 99% recall, and reducing approximate Top-K cost below the cost of the preceding matrix multiplication.

**Audience:**

Yes

**Audience Explanation:**

Yes. Top-K selection is a genuine bottleneck in modern ML systems — from sparse inference and mixture-of-experts routing to retrieval-augmented generation. The motivating statistic (Top-K taking 27x longer than the matmul on TPUv5e) is compelling and practically relevant.

Practitioners building sparse model architectures or MIPS systems on TPUs would directly benefit from this work. The automatic parameter selection (approx_top_k(array, K, recall_target)) interface is a practical improvement over Key et al.'s manual-tuning API.

**Claims And Evidence:**

Yes

**Claims Explanation:**

I think the claims are largely substantiated -- The theoretical results (Sections 6.1–6.2, Theorem 1) seem correct to me, and the Monte Carlo validation in Appendix A.2 is convincing.

There is one point where the abstract claims speedups "without sacrificing recall on real-world tasks," but all recall measurements in the paper are computed under the random placement model (elements uniformly distributed across buckets). The only "real-world" aspect is that the array sizes match practical workloads, this is a small concern because In actual LLM activations, top-K elements are not uniformly distributed.

**Requested Changes:**

It would be good to have one experiment that explicitly evaluates downstream task quality, as is done in Key et al (https://arxiv.org/pdf/2412.04358).

The compute cost for the Monte Carlo estimation is unclear, it would be good to have a discussion on this.

---

> ### Author Response · Authors · 2026-03-28
> **Thank you for your comments.**
>
> > There is one point where the abstract claims speedups "without sacrificing recall on real-world tasks," but all recall measurements in the paper are computed under the random placement model (elements uniformly distributed across buckets). The only "real-world" aspect is that the array sizes match practical workloads, this is a small concern because In actual LLM activations, top-K elements are not uniformly distributed.
>
> > It would be good to have one experiment that explicitly evaluates downstream task quality, as is done in Key et al (https://arxiv.org/pdf/2412.04358).
>
> We agree that this phrasing is imprecise and have revised the abstract to avoid any potential confusion. The "real-world" aspect of our evaluation is indeed the array sizes and values of K, which match practical workloads, and we believe that the recall is maintained in practice for two reasons.
>
> 1. Our implementation uses a stride-based bucket assignment, the same approach used by both Chern et al. and Key et al. In this scheme, elements separated by a fixed stride are grouped into the same bucket, giving perfect recall of 1 for sorted sequences. In LLM activations, even if the top-K elements are clustered together (e.g., in attention), stride-based assignment spreads adjacent elements into different buckets. The only adversarial case, both for our algorithm and those of Chern et al. and Key et al., is where the top-K elements fall at positions that are exact multiples of the stride. Given the wide usage of the Chern et al. algorithm, we believe that this adversarial case is rarely observed in practice. In the unlikely event such correlations are suspected, mitigations such as a global or chunk-level random shuffle, or choosing a different stride will be effective.
> 2. Key et al. explicitly evaluates downstream task quality for the same generalized algorithm (K' > 1) as ours, across several tasks. Since the quality of the algorithm is independent of the specific implementation, for the same algorithm, these experiments directly confirm the downstream task utility of the proposed approach.
>
> > The compute cost for the Monte Carlo estimation is unclear, it would be good to have a discussion on this.
>
>
> The Monte Carlo estimation is used solely for offline parameter selection that fixes the algorithm parameters during compilation. We have added a discussion of the compute cost to Appendix A.8.2. Here is an excerpt:
>
> *The parameter sweep evaluates configurations in descending order of bucket count for each value of K′, terminating early once the recall target is no longer met. This works since recall decreases monotonically with fewer buckets. For each configuration evaluated, we use an adaptive sampling procedure that draws hypergeometric samples until the recall estimate is within ±0.005 at 3-sigma confidence. To give a sense of the cost, we evaluated eight representative configurations spanning array sizes from 16k to 917k and K from 128 to 3360 at 95% recall target; the parameter sweep evaluated between a few dozen and up to 115
> configurations per (N, K, recall_target) combination, drawing between 930k and 5M hypergeometric samples in total, and completed in under a second on an AMD Ryzen 7 3700X desktop CPU. This cost is negligible relative to the typical compilation times for LLMs, which range from a few minutes to several hours. In practice, the selected parameters can be cached and reused across all calls with the same configuration, which is common in LLMs where repeated identical blocks of layers dominate the architecture. The implementation has not
> been optimized; straightforward improvements such as binary search over bucket counts or sharing random samples across configurations could further reduce the cost.*

---

> > ### Comment · Reviewer_ZuW6 · 2026-04-16
> >
> > Thank you for your response, my comments have been sufficiently addressed.

---

### Review · Reviewer_wW7M · 2026-03-14

**Summary Of Contributions:**

The paper addresses the performance bottleneck of Top-K selection in machine learning tasks on accelerators by generalizing the two-stage approximate Top-K algorithm originally proposed by Chern et al. (2022) . While previous methods selected only the top-1 element from each partition in the first stage, the authors propose selecting the top $K'$ elements, demonstrating that $K' > 1$ with fewer partitions more effectively reduces the input size for the expensive second stage while maintaining the same expected recall. Their contributions include a new closed-form expression for expected recall, a tighter theoretical bound for the original $K'=1$ setting, and optimized TPU implementations, including a matmul-fused version that merges the first Top-K stage with matrix multiplication. Empirical results show that their approach achieves reduction in latency over existing implementations.

**Audience:**

Yes

**Audience Explanation:**

The problem of Top-K selection in machine learning tasks on accelerators is of interest to the community.

**Claims And Evidence:**

No

**Claims Explanation:**

- Limited Hardware Scope: While the authors suggest the ideas apply to GPUs, the empirical evaluation and specific implementations are focused exclusively on Cloud TPU. Performance on other architectures are not discussed. Given a larger number of AI/ML runs on GPUs, it is essential for the authors to benchmark on GPUs for being considered for a transactions paper.
- Performance Degradation at Small Scale: Because of the requirement for buckets to be multiples of 128, the algorithm actually performs worse than the baseline for smaller values of $K$.
- Assumption of Random Distribution: The theoretical analysis for expected recall assumes that the top-$K$ elements are placed randomly and uniformly in the input array. In real-world data where top values might be clustered or structured, the actual recall might differ significantly from the predicted expected recall.
- No Comparison to Exact Algorithms: The authors explicitly state they do not compare their results with exact Top-$K$ algorithms for GPUs, arguing that their framework is intended to transform such algorithms rather than compete with them. This limits the scope of the work.
- No wall-clock latency improvements for language models: Authors have not made any effort to demonstrate to what extent their proposed methodology can provide benefits in practical scenarios with state-of-the-art language models.

**Requested Changes:**

Please address the issues raised above.

---

> ### Author Response · Authors · 2026-03-28
> **Thank you for your comments - part 1.**
>
> > Limited Hardware Scope: While the authors suggest the ideas apply to GPUs, the empirical evaluation and specific implementations are focused exclusively on Cloud TPU. Performance on other architectures are not discussed. Given a larger number of AI/ML runs on GPUs, it is essential for the authors to benchmark on GPUs for being considered for a transactions paper.
>
> We acknowledge that GPUs are an important hardware for AI/ML workloads. We note, however, that TPUs are also extensively used in the industry, including by several frontier AI labs such as Anthropic and Google DeepMind. We respectfully disagree that work that improves implementations on hardware such as TPUs are not worthy of a transactions paper irrespective of the technical merit.
>
> That said, most of our contributions generalize beyond TPUs. Our performance modeling framework (Section 2.3) explicitly characterizes A100 and H100 alongside TPUs, and the theoretical results in Section 6.2 and empirical results in Section 7.1 are hardware-agnostic.
> Together, they allow us to reason about the optimal choices of K’ for any hardware given the equivalent values for Table 1. In fact, our paper provides the necessary building blocks for someone to build a complete high-quality implementation on any target hardware. The only hardware-specific contribution in our paper is the kernel itself, which necessarily depends on the target platform. The choice of target platform is a natural and reasonable scope decision that reflects the authors’ deployment context, and does not diminish the generality of the contributions.
>
> Furthermore, writing a high-quality kernel for a new platform requires hardware-specific expertise and substantial engineering effort. It is often beyond the scope of a single academic paper to provide implementations for multiple hardware. Most papers, including widely recognized ones such as FlashAttention, focus on a single platform. Chern et al. (TPUs) and Key et al. (GPUs) are further examples within this specific line of work.
>
>
> $\space$
>
> > Performance Degradation at Small Scale: Because of the requirement for buckets to be multiples of 128, the algorithm actually performs worse than the baseline for smaller values of $K$.
>
> The perceived degradation arises from how Figure 4 was presented. The original figure isolated the *additional gains* from K’>1 over K’=1 baseline, and the cells with factors smaller than one indicated configurations where K’>1 does not improve over K’=1 – not that the algorithm underperforms Chern et al. Our *generalized* algorithm selects the best K’ from {1, 2, …}, as described in Appendix 8, and therefore always performs at least as well as the K’=1 baseline by construction. We have revised Figure 4 to show the reduction factor of our algorithm relative to Chern et al. when optimally selecting K′.
>
> That K’=1 is optimal for small K is an expected consequence of our implementation constraint. We require the number of buckets to be a multiple of 128 for simplicity and performance, which causes bucket count to be rounded **up** to 128 in the small-K regime, sometimes resulting in more output elements in K’ > 1 configurations than K’ = 1 baseline. Such constraints are common in performant implementations.
>
> Furthermore, optimizing for this small K regime isn’t worthwhile for two reasons. The K’=1 configurations appear optimal only when B is of the order of 128, where rounding to the nearest higher multiple of 128 could cause a large relative increase in the number of buckets that outweighs the algorithmic gains from K’ > 1. Our bound for K’=1 gives $B \approx \frac{K}{2(1-r)}$, so for a recall target of $99$%, $B \approx 50K$, which is of order 128 only when $K \sim 2$, which are pretty extreme configurations and a narrow regime. In this regime, even after rounding, the total number of elements entering the second stage  – $B\times K’$ – remains in the low hundreds, making the second stage trivially fast regardless of $K’$.

---

> ### Author Response · Authors · 2026-03-28
> **Thank you for your comments - part 2.**
>
> > Assumption of Random Distribution: The theoretical analysis for expected recall assumes that the top-$K$ elements are placed randomly and uniformly in the input array. In real-world data where top values might be clustered or structured, the actual recall might differ significantly from the predicted expected recall.
>
> Our implementation uses a stride-based bucket assignment, the same approach used by both Chern et al. and Key et al. In this scheme, elements separated by a fixed stride are grouped into the same bucket, giving perfect recall of 1 for sorted sequences. In LLM activations, even if the top-K elements are clustered together (e.g., in attention), stride-based assignment spreads adjacent elements into different buckets. The only adversarial case, both for our algorithm and those of Chern et al. and Key et al., is where the top-K elements fall at positions that are exact multiples of the stride. Given the wide usage of the Chern et al. algorithm, we believe that this adversarial case is rarely observed in practice. In the unlikely event such correlations are suspected, mitigations such as a global or chunk-level random shuffle, or choosing a different stride will be effective.
>
> $\space$
>
> > No Comparison to Exact Algorithms: The authors explicitly state they do not compare their results with exact Top-$K$ algorithms for GPUs, arguing that their framework is intended to transform such algorithms rather than compete with them. This limits the scope of the work.
>
> Our algorithm is a generic wrapper that transforms any exact Top-K algorithm into an approximate one by adding a lightweight first stage. It therefore does not compete with exact algorithms; it subsumes and generalizes them. Since the first stage only reduces the input size for the second stage, our algorithm is always at least as fast as whatever exact algorithm is used in the second stage, at any given recall target. Furthermore, the motivation for approx. Top-K algorithms such as Chern et al. is precisely that exact Top-K is too slow for practical use on accelerators, and we’re directly improving upon this widely deployed baseline.
>
> $\space$
>
> > No wall-clock latency improvements for language models: Authors have not made any effort to demonstrate to what extent their proposed methodology can provide benefits in practical scenarios with state-of-the-art language models.
>
> We demonstrate a concrete training scenario in the introduction: identifying sparse activations in the feedforward blocks of Gemma 2 9B, where exact Top-$K$ takes $27\times$ longer than the preceding matmul and the existing approximate algorithm still takes $9\times$ longer. Several Top-$K$-based sparsity approaches have been proposed in the literature, both for FFN activations (e.g., arxiv:2402.09360) and attention mechanisms (e.g., arxiv:2106.06899), and Top-$K$ is often a significant and non-trivial bottleneck. Our algorithm directly addresses this bottleneck, the speedups we demonstrate translate into significant end-to-end gains in such works.

---

> ### Comment · Reviewer_wW7M · 2026-04-13
> **Follow up on rebuttal**
>
> Thanks to the authors for their rebuttal.
>
> I checked the revised manuscript, Section 2.3, the authors state: "In this section, we restrict ourselves to TPUs and focus on three key subsystems." I do not see any mention of how their modeling framework characterizes A100/H100. The authors should state that explicitly and also include suitable references for substantiating their claim in the rebuttal.
>
> Regarding wall-clock timing results, while individual gains seem plausible, end to end benchmarking with at least one dataset/model combination is essential. Already the manuscript is restricted to TPUs. The authors should make an attempt to demonstrate actual gains in practice at least for 1 dataset/model on TPUs.
>
> My other comments have been sufficiently addressed.

---

> ### Author Response · Authors · 2026-04-18
>
> > I checked the revised manuscript, Section 2.3, the authors state: "In this section, we restrict ourselves to TPUs and focus on three key subsystems." I do not see any mention of how their modeling framework characterizes A100/H100. The authors should state that explicitly and also include suitable references for substantiating their claim in the rebuttal.
>
> Section 2.1 and 2.2 introduce the key concepts underlying both GPU and TPU architectures. In Section 2.3, we restrict the discussion to TPUs primarily for the ease of exposition as it's easier to follow with a concrete hardware and also sets up context and terminology for subsequent TPU-specific discussions (e.g., VPU). However, the same analysis applies to GPUs and Table 1 provides the values for the hardware constants ($\beta$, $\gamma$, $\pi$) for A100 and H100 GPUs. We will explicitly note this in our next revision.
>
> > Regarding wall-clock timing results, while individual gains seem plausible, end to end benchmarking with at least one dataset/model combination is essential. Already the manuscript is restricted to TPUs. The authors should make an attempt to demonstrate actual gains in practice at least for 1 dataset/model on TPUs.
>
> We benchmarked a non-gated MLP variant of Gemma 2 9B with SquaredReLU activations for inducing sparsity [1-3]. The model uses an intermediate dimension of 24,576, adjusted to roughly match the parameters of the gated model, while keeping all other hyperparameters unchanged. We use per-rank batch size 8 and sequence length of 1024, and apply Top-K to find the top 2% of FFN activations (K = 512 / 24,576 ≈ 2.08%) at a 95% recall target. This setup matches the configuration used in our introduction section.
>
> When measuring the total cost of each isolated residual block (forward + backward), a dense MLP residual block takes 33ms and attention block takes 16ms. With Chern et al.'s Top-K, sparse MLP blocks takes 89ms, which is nearly 2.7x the dense MLP block cost, which makes Top-K operation dominate the MLP block as well as the entire transformer block (MLP + attn ~ 50ms). Our algorithm raises the sparse MLP time to only 38ms, a modest +5ms overhead. This improvement results in a 1.91x end-to-end training throughput gain over Chern et al.
>
> [1] ReLU^2 Wins: Discovering Efficient Activation Functions for Sparse LLMs
>
> [2] ReLU Strikes Back: Exploiting Activation Sparsity in Large Language Models
>
> [3] HiRE: High Recall Approximate Top-k Estimation for Efficient LLM Inference

---

> > ### Comment · Reviewer_wW7M · 2026-04-20
> > **Follow up on rebuttal**
> >
> > Thanks for the rebuttal.
> > The authors should add their benchmark result to the revised manuscript, along with the clarification that Sections 2.1 and 2.2 are for GPUs and TPUs and Section 2.3 is restricted to TPUs.

---

> > > ### Author Response · Authors · 2026-04-23
> > >
> > > Sections 2.1 and 2.2 already discusses both GPUs and TPUs explicitly. We have added a clarification in Section 2.3 to highlight that the analysis applies to GPUs too. Due to space constraints, we've added the detailed benchmark result to the Appendix and a short reference to it in the introduction.

---

### Decision · Action_Editor_NV2Y · 2026-04-21

**Recommendation:** Accept with minor revision

**Additional Comments:**

This submission received positive reviews overall, and after the rebuttal and discussion all reviewers ended up recommending acceptance or leaning acceptance. The paper addresses an important practical problem, provides a meaningful improvement over prior two-stage approximate Top-$K$ methods, and combines solid theoretical analysis with strong empirical results on TPUv5e.

The remaining issues are minor and mainly concern presentation and positioning. The final version should clearly separate the hardware-agnostic contributions from the TPU-specific implementation details, incorporate the additional end-to-end benchmark discussed during rebuttal, and keep the discussion of concurrent work and limitations clear.

**Audience:**

Yes

**Audience Explanation:**

This paper studies efficient approximate Top-$K$ selection, which is an important systems bottleneck in modern machine learning workloads. The topic should be of interest to members of the TMLR audience working on efficient ML systems, sparse computation, retrieval, and accelerator-aware algorithm design.

**Claims And Evidence:**

Yes

**Claims Explanation:**

The paper provides both theoretical and empirical support for its main claims. The reviewers found the theoretical analysis of the generalized two-stage approximate Top-$K$ algorithm to be sound, including the expected recall expression and the improved bound for the $K^′=1$ setting. The empirical results on TPUv5e also provide clear evidence of substantial speedups over the existing baseline.